# SPACE: SPike-Aware Consistency Enhancement for Test-Time Adaptation in Spiking Neural Networks

**Xinyu Luo    Kecheng Chen    Pao-Sheng Vincent Sun    Chris Xing Tian**
**Arindam Basu    Haoliang Li**[*]
Department of Electrical Engineering, City University of Hong Kong
{xinyuluo8-c, kechechen3-c, vincesun-c, xingtian4-c}@my.cityu.edu.hk
{arinbasu, haoliali}@cityu.edu.hk

## Abstract

Spiking Neural Networks (SNNs), as a biologically plausible alternative to Artificial Neural Networks (ANNs), have demonstrated advantages in terms of energy efficiency, temporal processing, and biological plausibility. However, SNNs are highly sensitive to distribution shifts, which can significantly degrade their performance in real-world scenarios. Traditional test-time adaptation (TTA) methods designed for ANNs often fail to address the unique computational dynamics of SNNs, such as sparsity and temporal spiking behavior. To address these challenges, we propose SPike-Aware Consistency Enhancement (SPACE), the first source-free and single-instance TTA method specifically designed for SNNs. SPACE leverages the inherent spike dynamics of SNNs to maximize the consistency of spike-behavior-based local feature maps across augmented versions of a single test sample, enabling robust adaptation without requiring source data. We evaluate SPACE on multiple datasets. Furthermore, SPACE exhibits robust generalization across diverse network architectures, consistently enhancing the performance of SNNs on CNNs, Transformer, and ConvLSTM architectures. Experimental results show that SPACE outperforms state-of-the-art ANN methods while maintaining lower computational cost, highlighting its effectiveness and robustness for SNNs in real-world settings. The code will be available at https://github.com/ethanxyluo/SPACE.

## 1 Introduction

Recent advancements in neuroscience-inspired computing have placed Spiking Neural Networks (SNNs) at the forefront as a biologically plausible alternative to traditional Artificial Neural Networks (ANNs). While ANNs have achieved remarkable success across domains, their reliance on dense, black-box architectures often limits interpretability and energy efficiency. In contrast, SNNs emulate the sparse, event-driven dynamics of biological neurons, offering advantages in computational efficiency, temporal processing, and explainability [13, 4, 44]. However, the very features that make SNNs appealing temporal coding and sparse spiking, also introduce fragility in dynamic real-world environments. Because SNNs encode information over time, they can be highly sensitive to shifts in the input distribution [2], which may induce substantial changes in layer-wise spiking activity and degrade downstream performance. At the same time, the historical SNN–ANN accuracy gap on standard vision benchmarks has narrowed considerably [58, 52], foregrounding generalization and robustness as primary objectives for practical SNN deployment [10, 51]. In application domains central to SNNs, event-driven sensing and low-power edge inference, the ability to remain robust to distribution shift is thus a deployment prerequisite rather than a post-hoc enhancement.

---

[*]Corresponding author

39th Conference on Neural Information Processing Systems (NeurIPS 2025).

In real-world deployments, it is common for the training (*a.k.a.* source) and test (*a.k.a.* target) distributions to differ owing to changes in lighting, background, sensor noise, or environmental conditions. Such domain shifts are known to impair model performance [18, 30]. Prior studies have specifically noted robustness challenges of SNNs under distribution shift [38, 27, 23], but remain hard to deploy online at scale. Mei et al. [38] perform gradual, unsupervised adaptation via teacher–student self-training across intermediate domains, which requires iterative weight updates, pseudo-label calibration, and multiple passes during deployment with assumptions that clash with single-sample, low-latency edge constraints. Karilanova et al. [27] address only temporal-resolution shifts via parameter remapping that presumes a known source–target sampling ratio and synchronized binning, offering no per-instance online adaptation and limited coverage of broader appearance/sensor shifts. To address the limitations, we adopt SNN-specific test-time adaptation (TTA) to meet practical deployment constraints.

Existing TTA approaches often rely on source data or batched target samples, e.g., using source-related proxy tasks [48], source statistics [40], or optimization over a batch of test inputs [49, 54, 34]. Such assumptions are frequently impractical due to privacy, regulatory, and on-device resource constraints. We therefore focus on the source-free, single-instance regime, where the model adapts online using only the current test sample and without access to source data or target batches. Note that several influential ANN-oriented TTA methods fall outside this regime. For example, SHOT [37] requires mini-batches of target data and multi-pass self-training in a static feature space while freezing a standalone classifier head; in SNNs, classification is tightly coupled to time-evolving spiking dynamics, leaving no stationary head to freeze and rendering distance-based pseudo-labeling insensitive to temporal structure. Similarly, TAST [25] relies on class-prior regularization and batch-level target statistics and is typically optimized over target mini-batches, assuming misaligned with sparse, discrete spike trains and incompatible with the single-instance constraint.

Within the truly source-free, single-instance family, MEMO [56] and SITA [28] are representative but face fundamental limitations for SNNs. MEMO enforces output-consistency by minimizing the entropy of the averaged prediction over augmented views; however, output probabilities capture only coarse semantics and provide weak control over the fine-grained temporal spiking dynamics that govern internal representations. SITA adapts batch-normalization (BN) statistics using test-time augmentations, yet many SNN architectures omit BN layers [55, 32]; even when present, sparse activations limit the effect of BN-moment shifts on spike generation. These gaps motivate an SNN-specific TTA framework that explicitly engages temporal spiking dynamics, avoids reliance on BN, and respects the source-free, single-instance constraints to improve robustness under domain shift.

In this work, we propose a novel method named **SP**ike-**A**ware **C**onsistency **E**nhancement (SPACE), which is the first source-free and single-instance TTA approach specifically designed for SNNs. SPACE performs adaptation using only a single test point without access to source data, making it particularly suitable for real-world scenarios where source data are unavailable or privacy-sensitive. By leveraging the inherent spike dynamics of SNNs, SPACE maximizes the consistency in spike-behavior-based local feature maps across augmented samples of the same input, ensuring robust and efficient adaptation in deep SNNs. Unlike existing TTA methods, which often overlook the unique characteristics of SNNs and the demanding of single instance scenarios, SPACE directly exploits spike-based representations for adaptation. Our contributions can be summarized as follows

1. To the best of our knowledge, we are the first TTA method tailored for SNNs, which operates using only a single test sample. This addresses the unique challenges of adapting SNNs in scenarios where source data are unavailable, while maintaining efficiency and accuracy.

2. Our method introduces a consistency-driven optimization framework to enhance the similarity of spike-behavior-based local feature maps across augmented samples of the single point. By leveraging spike dynamics, this approach ensures robust adaptation in deep SNNs.

3. To assess the effectiveness and robustness of our proposed method, we perform extensive experiments across multiple benchmarks, including CIFAR-10, CIFAR-100 [31], Tiny-ImageNet, ImageNet [7], and the neuromorphic dataset DVS Gesture [1]. Furthermore, we validate the adaptability of our method across different model architectures, specifically testing on SNN-VGG [47, 29], SNN-ResNet [17, 32], Spike-driven Transformer V3 [52], and SNN-ConvLSTM [46]. The results demonstrate that our approach not only achieves consistent performance improvements across datasets but also generalizes well to different network structures, showcasing its broad applicability and robustness.

## 2 Related Work

### 2.1 SNN-based Deep Learning

Over the years, significant progress has been made in SNN-based deep learning. Many approaches primarily relied on converting pre-trained ANN models into SNNs, enabling SNNs to inherit the representational power of ANNs while benefiting from spike-based computations [9, 3, 22, 26]. However, such conversion methods often suffer from performance degradation due to differences in activation dynamics. To overcome this, direct training of SNNs using surrogate gradient methods has gained traction, enabling end-to-end optimization of spiking models [55, 39, 12, 8, 36]. Rathi et al. [41] proposed a hybrid method that combines conversion and surrogate gradient-based method, achieving state-of-the-art performance. These advancements have not only solved the challenge owing to the non-differentiable nature of spike functions and the the inherent complexity of temporal dynamics, but also led to the development of sophisticated SNN architectures, such as spiking convolutional networks [50, 32], spiking recurrent networks [53, 57], and spiking transformer networks [58, 52], which have denoted promising results in diverse tasks like image classification, speech recognition, and event-driven sensor data analysis.

### 2.2 Test-time Adaptation

Test-time adaptation (TTA) is a rapidly growing area in machine learning that tackles the challenge of adapting pre-trained models to unseen or shifted distributions during inference [37, 48, 49, 24]. Test-time training (TTT) [48] performs online updates to model parameters using supervised proxy task on the test data. However, the dependence on source data makes these methods impractical in scenarios where the source domain is unavailable during deployment due to privacy or storage constraints. To overcome source dependency, source-free TTA methods have been proposed, which perform adaptation using only test-time inputs. TENT [49] utilizes a straightforward yet effective entropy minimization approach to optimize batch normalization parameters during test time, without relying on any proxy task during training. Lee et al. [33] employs pseudo-labeling to adjust model predictions.

Although these methods avoid source data, many assume batch-level adaptation, relying on test-time statistics over multiple samples, which is an impractical requirement when only a single test point is available. MEMO [56] addresses the single-instance setting by minimizing entropy and enforcing prediction consistency across augmentations, thereby promoting augmentation-invariant semantics but offering limited control over temporal spiking dynamics under shift. SITA [28] adapts batch-normalization statistics using augmented views of the same instance; however, many SNNs lack BN [55, 32], and sparse activations diminish the effectiveness of moment updates, especially under severe corruptions. To overcome these limitations, we align spike patterns and temporal features across augmentations without relying on BN, yielding an SNN-specific TTA mechanism compatible with diverse architectures and truly single-sample operation, while offering a more stable and informative alignment signal for robustness.

### 2.3 Generalization of SNNs

Recent state-of-the-art SNNs [58, 52] achieve performance on clean data comparable to strong ANNs, shifting the research focus from in-distribution accuracy to generalization and robustness under real-world shifts. Several works have begun to examine these issues in SNNs [10, 15, 51]. Guo et al. [15] study Source-Free Domain Adaptation (SFDA) in a batch/epoch setting, where the model is updated after processing target-domain batches and typically assumes access to a pre-collected target dataset. In contrast, our work tackles TTA in a stricter, online regime: we adapt to each incoming sample individually and on the fly, without labels, source data, or target-set pre-collection. This per-sample update schedule reduces latency and memory footprint, making the approach directly applicable to streaming, autonomous deployments where SNNs are expected to excel.

# 3 Proposed Method

## 3.1 Preliminary

SSNNs emulate biological neurons by transmitting and encoding information through discrete spikes. Among various neuron models, the Leaky Integrate-and-Fire (LIF) neuron [6] is widely adopted for its balance between biological realism and computational efficiency. Its membrane potential dynamics follow

$$\tau_m \frac{U(t)}{dt} = -U(t) + RI(t),\tag{1}$$

where $U(t)$ represents the membrane potential of the neuron at time $t$, $\tau_m$ is the membrane time constant, $R$ denotes the input resistance and $I(t)$ is the input current received from pre-synaptic neurons or inputs. When $U(t)$ exceeds a predefined threshold $U_{th}$, the neuron emits a spike and its membrane potential is reset to a resting value, typically 0 or $U(t) - U_{th}$. Following [12], Equation 1 is discretized as

$$u_i^t = (1 - \frac{1}{\tau_m})u_i^{t-1} + \frac{1}{\tau_m}\sum_j w_{ij}o_j^t.\tag{2}$$

Here, $j$ is the index of pre-synaptic neurons, $o_j$ is the binary spike activation, and $w_{ij}$ stands for connections between pre- and post-neurons. Figure 1 shows the LIF model, depicting membrane potential evolution and spike generation in response to input spikes [11].

The spiking mechanism in SNNs introduces non-differentiability, making it difficult to apply standard gradient-based optimization methods for training or test-time adaptation. To address this, surrogate gradient methods approximate the non-differentiable spike function with a smooth surrogate during the backward pass. A simple approach is the shifted Heaviside step function, where the gradient of $o_j$, the spiking activation function of neuron $j$, with respect to $U$ is defined as

$$\frac{\partial o_j}{\partial U} \triangleq \begin{cases} 1, & \text{if } U \geq U_{th}, \\ 0, & \text{if } U < U_{th}. \end{cases}\tag{3}$$

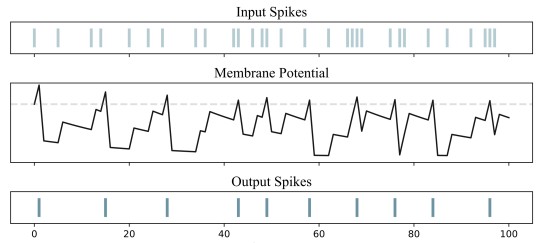

Figure 1: Illustration of spike and membrane potential dynamics in LIF neurons. When the membrane potential reaches the threshold, it is reset by subtracting the threshold value, triggering the neuron to fire a spike.

While this is not an exact analytical solution, it is nevertheless valid because a reset promptly occurs after a spike is generated when $U \geq U_{th}$.

These SNN foundational techniques underpin our method to enhance TTA by ensuring consistent spiking behavior across augmented samples.

## 3.2 SPike-Aware Consistency Enhancement

**Algorithm Design Overview** The proposed algorithm through spike-aware consistency enhancement (SPACE) for TTA is designed to improve test-time robustness of SNNs, as shown in Figure 2. We first generate an augmented batch from a single test sample using various augmentation techniques, introducing diversity while retaining the core characteristics of the original sample. This augmented batch is passed through the model to obtain local feature maps, represented by spike counts over time. Next, the model is adapted by maximizing the similarity across the feature maps of the augmented samples, promoting consistency in feature representations. Finally, the adapted model predicts the label of the original test sample, ensuring robust performance under test-time conditions. This approach enhances the model's ability to generalize to unseen test samples, particularly in shifted domains. The overall method is presented in Algorithm 1, and the details are introduced below.

**Spike-Aware Feature Maps** SPACE aims to adapt pre-trained SNN-based models $M_\theta$ (with parameters $\theta \in \Theta$) during inference to improve performance under distribution shifts, without requiring ground truth labels or access to source data. No specific training process is required, nor do we impose particular constraints on the model. The only assumptions are that $\theta$ can be adjusted

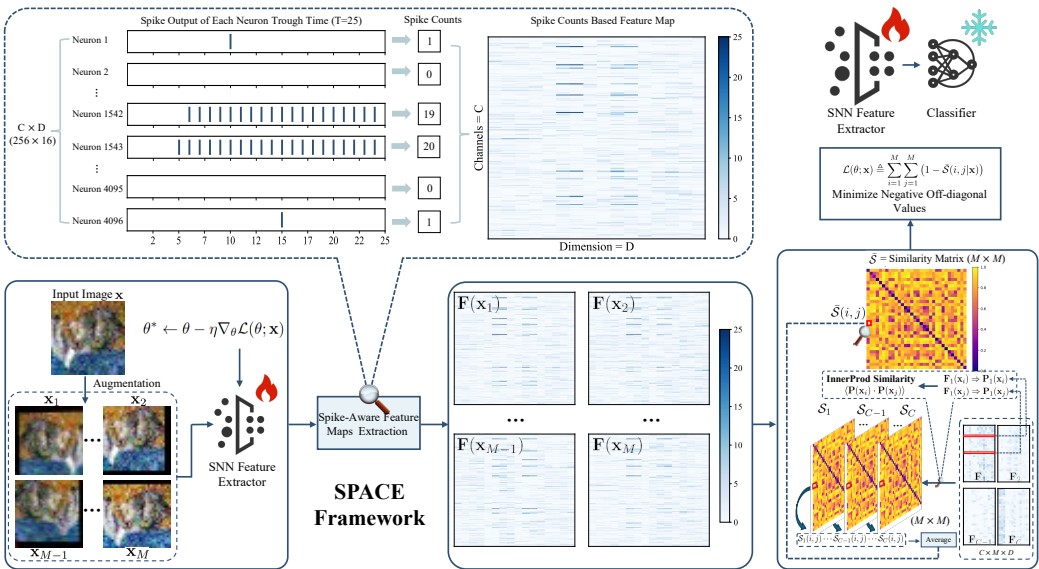

Figure 2: Overview of the **SP**ike-**A**ware **C**onsistency **E**nhancement (SPACE) framework for TTA in SNNs. The test sample is selected from CIFAR-10-C dataset [18] with Gaussian Noise corruption at level 5. The model here follows the VGG9 architecture. The process involves four main steps: 1) Generate an augmented batch from the single test sample. 2) Pass the augmented batch through the model to obtain local feature maps, represented by the spike counts over the processing time. 3) Adapt the model by maximizing the similarity across the local feature maps of the augmented samples. 4) Use the adapted model to predict the label of the original test sample.

and that the model produces intermediate feature maps $\mathbf{F}(\mathbf{x})$, which is differentiable with respect to $\theta$ and can be utilized for further analysis and adaptation. Here, $\mathbf{x} \in \mathcal{X}$ is a single test point that is presented to $M_\theta$. It is worth noting that this is a reasonable assumption, supported by a recent advance [43] that suggest gradient back-propagation can be implemented on spiking neuromorphic hardware. Concretely, we decompose the SNN into a spike-based feature extractor and a classifier, writing $M_\theta(\mathbf{x}) = C_{\theta_C}\big(E_{\theta_E}(\mathbf{x})\big)$ with $\theta = (\theta_E, \theta_C)$. Here, the extractor $E_{\theta_E}$ denotes the main body of the network (e.g., CNN/Transformer/ConvLSTM blocks), and the classifier $C_{\theta_C}$ consists of the last one or two readout layers.

Aligned with previous work [28, 56], we achieve test-time robustness by leveraging data augmentations and a self-supervised adaptation objective. In our work, a set of augmentation functions with different intensities selected from $\mathcal{A} \triangleq \{a_1, \ldots, a_N\}$ is applied to a single test input $\mathbf{x}$, forming an augmented batch of samples $\mathcal{B} = \{\mathbf{x}_1, \mathbf{x}_2, \ldots, \mathbf{x}_M\}, M \leq N$. Using this batch, we align the spike dynamics extracted from augmentations, ensuring that the model produces reliable and robust predictions in the presence of distribution shifts. SPACE differs from MEMO [56] by operating not on the conditional output $p_\theta(y|\mathbf{x})$ but on the spike dynamics of intermediate layers that determine temporal coding and semantic representations. Specifically, we observe the feature map $\mathbf{O}(\cdot) \in \mathbb{R}^{T \times C \times H \times W}$ at the deepest layer of $E_{\theta_E}$ that still retains spatial support. Here $T$ denotes the number of time steps, $C$ represents the number of channels, $H$ and $W$ are the spatial dimensions of the local feature maps. If the last pre-classifier feature map has shape with $H \times W > 1$, we use this layer; if global pooling or stride reduces it to $1 \times 1$, we use the penultimate layer of $E_{\theta_E}$. This choice balances semantic abstraction and alignment granularity, where using a $1 \times 1$ map yields too few alignment targets and risks overfitting, whereas too shallow a layer may overemphasize low-level variations.

To leverage the intermediate spike activity over the entire time window, let $\mathbf{O}(\mathbf{x}_i) \in \{0, 1\}$ denote binary spikes (1=fire, 0=silent, refers to subsection 3.1) at the chosen extractor layer for an augmented view $\mathbf{x}_i$. We aggregate spikes over time to form a spike-aware feature maps collection $\mathcal{F} = \{\mathbf{F}(\mathbf{x}_1), \mathbf{F}(\mathbf{x}_2), \ldots, \mathbf{F}(\mathbf{x}_M)\}$, where each $\mathbf{F}(\mathbf{x}_i) \in \mathbb{R}^{C \times D}$ represents the total spike counts of neurons across different channels, with $D = H \times W$ indicating the spatial dimensionality.

**Algorithm 1** SPACE for Test-Time Adaptation in SNNs

---

1: **Input:** Test sample $\mathbf{x}$, augmentation functions $\mathcal{A} = \{a_1, \ldots, a_N\}$, SNN model with extractor parameters $\theta_E$, and learning rate $\eta$
2: **Output:** Prediction $y^*$ via updated model
3: **Step 1: Generate augmented batch**
4: Initialize $\mathcal{B} = \emptyset$
5: **for** $a_k$ in a subset $\{a_1, \ldots, a_M\}$ of $\mathcal{A}$ **do**
6: $\quad \mathbf{x}_k = a_k(\mathbf{x})$, add $\mathbf{x}_k$ to $\mathcal{B}$
7: **end for**
8: **Step 2: Extract feature maps**
9: Pass $\mathcal{B}$ through the SNN feature extractor, obtain $\mathcal{F} = \{\mathbf{F}(\mathbf{x}_1), \ldots, \mathbf{F}(\mathbf{x}_M)\}$
10: **Step 3: Compute similarity**
11: **for** each pair of $(\mathbf{F}(\mathbf{x}_i), \mathbf{F}(\mathbf{x}_j))$ in $\mathcal{F}$ **do**
12: $\quad$ Compute similarity $\bar{\mathcal{S}}(i, j|\mathbf{x})$ using Equation 4
13: **end for**
14: **Step 4: Update model parameters**
15: Compute loss $\mathcal{L}(\theta; \mathbf{x})$ using Equation 5, update $\theta_E$ via SGD with learning rate $\eta$
16: **Step 5: Predict**
17: $y^* \leftarrow \arg\max_y p_{\theta_E^*}(y|\mathbf{x})$
18: **Return:** Prediction result $y^*$

---

We choose total spike counts for three main reasons: 1) SNNs often run for tens to thousands of steps [45, 16, 32, 41], making stepwise matching redundant and costly. 2) Aggregation smooths the loss landscape: if spikes jitter in time but counts match, the loss remains unchanged, so gradients issue a clear directive: make this neuron more or less excitable for this input, producing stable, macro-level alignment of core feature activations across the augmented views. 3) distribution shift primarily manifests as changes in spiking rates and spatial activation support in intermediate layers; $\mathbf{F}(\cdot)$ provides a direct, informative summary of these changes that output probabilities cannot capture.

**Feature Maps Alignment**  Our goal is to make the model focus on invariant, task-relevant features rather than noise or minor transformations introduced by augmentations or distribution shift. If the feature maps of augmented views are highly similar, the model becomes less sensitive at test time and generalizes better to unseen, unlabeled samples. Concretely, given the spike-aware feature map $\mathbf{F}(\mathbf{x}_i) \in \mathbb{R}^{C \times D}$ of an augmented view $\mathbf{x}_i$, we compute channel-wise local vectors $\mathbf{F}_c(\mathbf{x}_i) \in \mathbb{R}^D$ and normalize them across spatial locations with a softmax to obtain a probability distribution $\mathbf{P}_c(\mathbf{x}_i) \in \Delta^{D-1}$. This normalization emphasizes salient regions and suppresses noise, yielding a stable target distribution. We then measure similarity between two augmented views $\mathbf{x}_i$ and $\mathbf{x}_j$ by the average channel-wise inner product

$$\bar{\mathcal{S}}(i, j \mid \mathbf{x}) = \frac{1}{C} \sum_{c=1}^{C} \sum_{d=1}^{D} \mathbf{P}_{c,d}(\mathbf{x}_i) \, \mathbf{P}_{c,d}(\mathbf{x}_j), \tag{4}$$

which lies in $[0, 1]$ and attains 1 if and only if the per-channel spatial distributions match exactly. Based on this, we adapt the extractor parameters by minimizing the loss function

$$\mathcal{L}(\theta_E; \mathbf{x}) \triangleq \sum_{1 \le j < i \le M} \left(1 - \bar{\mathcal{S}}(i, j \mid \mathbf{x})\right), \tag{5}$$

using **a single SGD step** with learning rate $\eta$. Minimizing $\mathcal{L}$ enforces consistency across augmentations, regularizing the model to learn stable, robust representations and preventing overfitting to any specific view. Finally, predictions are obtained with the adapted model.

**Theoretical Motivation**  Our objective instantiates consistency regularization at inference time: under semantics-preserving, bounded-severity augmentations (e.g., AugMix [19]), we enforce invariance in internal representations across transformations. Information-theoretically, aligning $\mathbf{F}$ (and thus $\mathbf{P}$) suppresses augmentation-specific variability while preserving identity-relevant content, approximating an information bottleneck that favors invariant features; geometrically, when perturbations push a sample off the learned manifold, adaptation pulls it toward the local manifold spanned

Table 1: Performance of different model architecture under domain shift. Values are Top-1 accuracy (%); Acc. Loss = Clean - Shifted. "Shifted" denotes evaluation on shifted variants of each dataset; exact configuration described in Appendix C. SNN and ANN models share backbone and scale and are trained with the same protocol.

| Architecture | Dataset | ANN | | | SNN | | |
|---|---|---|---|---|---|---|---|
| | | Clean | Shifted | Acc. Loss | Clean | Shifted | Acc. Loss |
| VGG9 | CIFAR-10 | 90.53 | 77.25 | **13.28** | 90.61 | 67.86 | **22.74** |
| VGG11 | CIFAR-100 | 70.86 | 48.32 | **22.54** | 69.50 | 41.35 | **28.14** |
| VGG11 | Tiny-ImageNet | 61.24 | 19.76 | **41.48** | 58.36 | 14.56 | **43.80** |
| Transformer | ImageNet | 85.16 | 75.49 | **9.67** | 79.60 | 67.64 | **11.96** |

by its augmented views, effectively denoising the representation. In SNNs, shifts often manifest as event-rate changes and timing jitter that nudge membrane potentials $U$ around the threshold $U_{th}$, making spike generation highly sensitive; by aligning time-resolved features across augmentations, we reduce the variance of $U$ trajectories and spike counts, enlarge the effective margin $|U - U_{th}|$, and stabilize and spike timing dynamics. Consequently, robustness arises from coupling semantic consistency with temporal stabilization within the SNN-TTA framework.

We also explored richer feature relations via kernel embeddings and Maximum Mean Discrepancy (MMD) (Appendix B). In practice, the above linear-kernel objective already captures the essential distributional overlap with lower variance and cost, and stronger kernels did not bring noticeable gains while adding overhead. More discussion is presented in Appendix A.

## 4 Experimental Results

**Datasets and Models** The experiments were conducted on six benchmark datasets for out-of-distribution generalization: CIFAR-10-C, CIFAR-100-C, Tiny-ImageNet-C [18], ImageNet-V2 [42], ImageNet-R [20], and ImageNet-A [21]. The first three datasets include 15 corruption types across four categories and five severity levels, while the three ImageNet-based datasets capture real-world domain shifts, such as distributional discrepancies and semantic variations. Additionally, we evaluated on the neuromorphic dataset DVS Gesture-C [23], which introduces six corruption types. Four backbone models were used: 1) SNN-VGG with Batch-Normalization Through Time (BNTT)[29], using 25 inference steps; 2) SNN-ResNet with 30 inference steps[32]; 3) Spike-driven Transformer V3 (19M parameters) with 4 inference steps [52]; and 4) SNN-ConvLSTM with 32 inference steps. All models were pre-trained on their respective source domains to ensure fair evaluation on datasets with domain shifts.

**Compared Methods** We compared our approach against models pre-trained on clean datasets and evaluated without any test-time adaptation (henceforth referred to as No Adapt), which serves as a lower-bound reference for robustness under distribution shift. Given the limited availability of TTA methods specifically designed for SNNs, we selected two representative traditional TTA methods, MEMO [56] and SITA [28], as baselines. These methods represent the current state-of-the-art for source-free and single-instance TTA, respectively. MEMO enforces output consistency across augmented samples, making it inherently compatible with SNNs by directly adjusting their final outputs. SITA, originally designed for models with BN layers, was adapted for SNN-VGG with BNTT by updating the BN layer parameters during TTA. Additionally, we included RoTTA [54] and the recent DeYO [34] as baselines, despite their reliance on batch data. To ensure a fair comparison, we applied the same augmentation strategy as SPACE to construct a pseudo-batch for each sample.

**Implementation details** To maximize the effectiveness of our proposed method, all experiments conducted on CIFAR-10-C, CIFAR-100-C, Tiny-ImageNet-C , and DVS Gesture-C were performed with the highest severity level (level=5). In line with our setting and practical scenarios, the test batch size for all experiments was fixed to 1, and the model was updated only once per test sample. Following MEMO's experimental setup, we applied AugMix [19] augmentation to generate diverse augmented samples, using a batch size of 32 for CIFAR-10-C, CIFAR-100-C, Tiny-ImageNet-C, and DVS Gesture-C, and 64 for the ImageNet series. Please refer to Appendix C for more details.

Table 2: Performance comparison on CIFAR-10-C, CIFAR-100-C, and Tiny-ImageNet-C with CNN-based SNN models regarding **Accuracy** (%). The **bold** number indicates the best result.

| Method | Noise | | | Blur | | | | Weather | | | | Digital | | | | Average Acc. |
|---|---|---|---|---|---|---|---|---|---|---|---|---|---|---|---|---|
| | Gauss. | Shot | Impl. | Defoc. | Glass | Motion | Zoom | Snow | Fog | Frost | Brit. | Contr. | Elas. | Pix. | JPEG | |
| | | | | | | *CIFAR-10-C, SNN-VGG9* | | | | | | | | | | |
| No Adapt | 72.38 | 74.70 | 58.57 | 63.05 | 63.96 | 64.44 | 71.33 | 76.32 | 43.57 | 75.72 | 82.44 | 22.54 | 75.01 | 70.25 | 84.28 | 66.57 |
| RoTTA | 73.60 | 75.49 | 61.61 | 66.08 | 67.53 | 67.75 | 72.40 | 74.34 | 51.54 | 76.35 | 80.98 | 23.01 | 74.23 | 68.60 | 83.29 | 67.79 |
| DeYO | 73.97 | 76.16 | 62.45 | 66.29 | 67.37 | 67.60 | 73.52 | 74.90 | 50.68 | 76.23 | 81.30 | 20.90 | 75.19 | 70.44 | 82.85 | 67.99 |
| SITA | 73.06 | 74.15 | 58.41 | 62.94 | 64.21 | 64.36 | 70.72 | 76.67 | 43.40 | 75.24 | 82.51 | 22.02 | 74.74 | 69.62 | 84.16 | 66.41 |
| MEMO | 77.73 | **79.50** | 65.74 | 65.61 | 67.50 | 66.24 | 72.61 | 77.38 | 45.47 | 78.64 | 83.05 | 23.83 | **76.45** | 73.25 | **85.05** | 69.20 |
| SPACE (ours) | **77.98** | 79.34 | **69.41** | **71.59** | **67.76** | **72.14** | **74.67** | **78.43** | **52.80** | **79.59** | **83.22** | **23.85** | 75.49 | **76.24** | 82.88 | **71.03** |
| | | | | | | *CIFAR-10-C, SNN-ResNet11* | | | | | | | | | | |
| No Adapt | 72.88 | 74.49 | 49.59 | 67.77 | 63.53 | 65.14 | 73.06 | 78.11 | 42.84 | 73.12 | 81.71 | 12.72 | 74.88 | 77.08 | 82.47 | 65.96 |
| RoTTA | 73.42 | 75.66 | 49.93 | 67.89 | 63.79 | 64.88 | 72.75 | 78.14 | 44.34 | 73.59 | 80.36 | 12.90 | 74.68 | 78.25 | 83.12 | 66.25 |
| MEMO | 74.21 | 75.50 | 51.35 | 68.11 | 64.65 | 65.37 | **73.96** | 78.32 | 43.47 | 73.39 | **81.82** | 13.63 | 75.54 | 77.54 | 83.24 | 66.67 |
| SPACE (ours) | **75.90** | **77.81** | **56.85** | **68.66** | **67.64** | **66.02** | 73.82 | **78.74** | **45.47** | **74.04** | 81.28 | **14.94** | **76.79** | **80.32** | **83.96** | **68.15** |
| | | | | | | *CIFAR-100-C, SNN-VGG11* | | | | | | | | | | |
| No Adapt | 42.51 | 45.40 | 25.50 | 42.15 | 42.84 | 41.87 | 46.77 | 44.88 | 16.51 | 44.12 | 49.36 | 5.42 | 51.33 | 52.35 | 58.47 | 40.63 |
| RoTTA | 43.02 | 44.94 | 25.48 | 42.24 | 43.26 | 42.13 | 46.93 | 44.50 | 16.63 | 43.83 | 49.54 | 5.45 | 50.99 | 52.37 | 58.50 | 40.65 |
| DeYO | 43.00 | 45.45 | 25.64 | 42.02 | 42.88 | 42.47 | 46.61 | 44.97 | 16.79 | 43.82 | 49.30 | 5.45 | 51.34 | 52.85 | 58.45 | 40.74 |
| SITA | 43.01 | 44.83 | 25.48 | 42.23 | 43.29 | 42.11 | 46.93 | 44.49 | 16.65 | 43.84 | 49.53 | 5.56 | 50.99 | 52.48 | 58.71 | 40.68 |
| MEMO | 43.46 | 45.41 | 26.07 | 42.47 | 43.90 | 42.55 | 47.21 | 44.74 | 16.76 | 44.34 | 49.95 | 5.62 | 51.24 | 52.82 | **59.29** | 41.06 |
| SPACE (ours) | **44.71** | **46.73** | **27.99** | **43.35** | **44.47** | **43.46** | **48.42** | **45.70** | **17.62** | **44.98** | **50.41** | **5.69** | **51.45** | **53.81** | 58.96 | **41.58** |
| | | | | | | *Tiny-ImageNet-C, SNN-VGG11* | | | | | | | | | | |
| No Adapt | 12.43 | 15.20 | 8.74 | 7.58 | 6.50 | 14.48 | 13.59 | 15.58 | 5.27 | 18.06 | 15.01 | 1.32 | 20.53 | 31.62 | 31.15 | 14.47 |
| RoTTA | 12.67 | 15.34 | 8.89 | 7.55 | 6.56 | 14.57 | 13.94 | 15.75 | 5.21 | 18.06 | 14.82 | 1.38 | 21.53 | 32.21 | 31.25 | 14.65 |
| DeYO | 12.69 | 15.36 | 8.56 | 7.39 | 6.26 | 14.38 | 13.75 | 15.75 | 5.29 | 18.12 | 14.85 | 1.41 | 20.93 | 31.99 | 31.47 | 14.55 |
| SITA | 12.71 | 15.17 | 8.87 | 7.56 | 6.57 | 14.57 | 13.94 | 15.72 | 5.32 | 18.05 | 14.72 | 1.38 | 21.45 | 32.20 | 31.51 | 14.66 |
| MEMO | 13.64 | 16.45 | 9.51 | 7.51 | 6.60 | 14.38 | 13.92 | 15.97 | 4.91 | 17.83 | 14.74 | 1.44 | 21.00 | 31.39 | 30.82 | 14.67 |
| SPACE (ours) | **16.71** | **19.44** | **11.72** | **9.62** | **7.31** | **16.57** | **15.61** | **18.66** | **6.01** | **21.02** | **17.53** | **1.51** | **21.96** | **31.89** | **31.63** | **16.48** |

## 4.1 Evaluation of Robustness of SNNs under Domain Shift.

To further substantiate the observation that SNNs are particularly vulnerable to domain shifts, we conducted a controlled comparison between SNNs and ANNs that share the same backbone family and model scale, trained under identical protocols, and evaluated on shifted variants of standard benchmarks (details in Appendix C). As summarized in Table 1, SNNs consistently incur larger accuracy drops than their ANN counterparts across CIFAR-10, CIFAR-100, Tiny-ImageNet, and ImageNet. These empirical results motivate the development of robust, SNN-specific TTA strategies tailored to spiking computation.

## 4.2 Main Results

In this section, we compare SPACE's performance with four existing TTA methods. To demonstrate the broad applicability of the proposed method, we evaluate all methods on CIFAR-10-C, CIFAR-100-C, and Tiny-ImageNet-C datasets using SNN-VGG, on CIFAR-10-C with SNN-ResNet11, on ImageNet series with Spike-driven Transformer V3, and on DVS Gesture-C with SNN-ConvLSTM, to demonstrate the broad applicability of our proposed method.

**Performance Comparison on CNN-based SNN Models** The results on CIFAR-10-C, CIFAR-100-C, and Tiny-ImageNet-C at level 5 corruption, shown in Table 2, highlight the effectiveness of SPACE. Key observations include: **1) Optimal Performance:** SPACE consistently outperforms all baselines in average accuracy across all datasets under both VGG and ResNet architectures. **2) Broad Applicability:** SPACE demonstrates effectiveness across different CNN structures, showcasing its generalizability. **3) Robustness on Complex Datasets:** A particularly notable result is SPACE's performance on Tiny-ImageNet-C, where it surpasses all methods across every corruption type. This highlights its robustness in handling complex datasets like Tiny-ImageNet-C. **4) Vs. RoTTA:** RoTTA's memory bank cannot be effectively utilized in this single-sample setting, leading to limited performance gains. **5) Vs. SITA:** The SNN-ResNet does not include BN layers, rendering SITA inapplicable. Furthermore,

Table 3: Performance comparison on ImageNet-V2/R/A with Spike-driven Transformer V3 regarding **Accuracy** (%). The **bold** number indicates the best result.

| Method | Accuracy (%) | | | |
|---|---|---|---|---|
| | V2 | R | A | Avg. |
| No Adapt | 67.64 | 41.94 | 15.12 | 41.57 |
| RoTTA | 67.34 | 43.12 | 14.87 | 41.78 |
| DeYO | 67.92 | 41.42 | 15.42 | 41.59 |
| SITA | 67.21 | 42.05 | 13.92 | 41.06 |
| MEMO | 67.57 | 42.15 | 14.97 | 41.56 |
| SPACE (ours) | **68.82** | **45.07** | **16.22** | **43.37** |

Table 4: Performance comparison on DVS Gesture-C with ConvLSTM-based SNN model regarding **Accuracy** (%). The **bold** number indicates the best result.

| Method | Accuracy (%) | | | | | | |
|---|---|---|---|---|---|---|---|
| | DropPixel | DropEvent | RefractoryPeriod | TimeJitter | SpatialJitter | UniformNoise | Avg. |
| No Adapt | 53.03 | 54.55 | 36.74 | 31.06 | 92.05 | 91.67 | 59.85 |
| RoTTA | 55.38 | 56.54 | 37.50 | 30.68 | 91.62 | 92.05 | 60.68 |
| MEMO | 56.06 | 55.30 | 36.74 | 30.68 | 90.91 | 91.67 | 60.23 |
| SPACE (ours) | **57.20** | **57.95** | **52.27** | **43.18** | **92.42** | **92.42** | **65.91** |

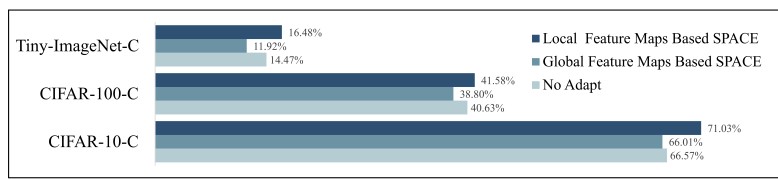
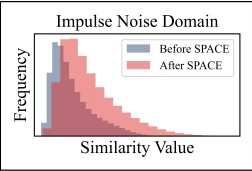

Figure 3: a) Local feature maps based SPACE outperforms global feature maps based SPACE. b) Similarity distribution of spike count feature maps before and after SPACE adaptation.

SITA performs worse than No Adapt, indicating its ineffectiveness for SNNs. This may be attributed to the spike-based transmission in SNNs: directly adjusting BN parameters without gradient guidance can disrupt spike information, leading to significant performance degradation. **6) Vs. DeYO:** DeYO relies on smooth probability outputs for confidence comparison, which is incompatible with SNN's discrete, temporal spike-based outputs, introducing significant errors. Also, DeYO only works normalization layer. **7) Vs. MEMO:** MEMO exhibits slightly better performance in a few specific cases, such as JPEG corruption on CIFAR-10-C (85.05% for MEMO vs. 82.88% for SPACE) and CIFAR-100-C (59.29% for MEMO vs. 58.96% for SPACE). The minor edge cases hint at MEMO's advantage under certain corruption types, yet they don't diminish SPACE's overall superiority.

**Performance Comparison on Spike-driven Transformer V3**   To explore the real-world impact of SPACE, we conducted tests on diverse datasets, including ImageNet-V2, ImageNet-R, and ImageNet-A. Furthermore, to evaluate its adaptability in deeper SNN architectures, we tested SPACE on the state-of-the-art Spike-driven Transformer V3. As shown in Table 3, SPACE consistently outperforms all baselines, achieving at least 1.59% improvement, which demonstrates its robustness in real-world scenarios and its effectiveness in advanced SNN models.

**Performance Comparison on ConvLSTM-based SNN Model**   SNNs are well-suited to event-based neuromorphic data, thanks to their temporal dynamics and energy efficiency. The DVS-Gesture dataset [1] is an ideal benchmark to evaluate the performance of SPACE on neuromorphic datasets, highlighting its compatibility with dynamic vision tasks and data with temporal structures. To further explore potential network architectures, we incorporated ConvLSTM. BN-dependent SITA and DeYO is not suitable for this architecture. As shown in Table 4, SPACE achieves significantly better results compared to existing methods, demonstrating its effectiveness and adaptability.

### 4.3   Ablation Studies

**Local vs. Global Feature Map Similarity**   *Definitions.* Let "global feature maps" refers to treating the entire feature map of size $C \times H \times W$, as a single entity and measuring similarity on this full representation. Let the "local feature map" considers each spatial position in the $H \times W$ dimension independently, which is adopted to calculate the similarity and averaging across the $C$ channels in the main paper.

We compared the performance of local versus global feature map similarity for our proposed method. The results in Figure 3a clearly illustrate the decisive advantage of using similarity of local feature maps over global feature maps. While the global approach does not account for variability in neuron activations across channels, the local method isolates the contributions of active neurons within each channel, offering a more stable and meaningful measure of feature map consistency. As shown in Figure 3a, the local method yields a substantial accuracy improvement, whereas the global approach negatively impacts performance across all three datasets. This demonstrates that

Table 5: Comparison of computational cost on CIFAR-10-C with SNN-VGG9 model.

| Method | RoTTA | DeYO | SITA | MEMO | SPACE |
|---|---|---|---|---|---|
| GFLOPs | 160.67 | 318.07 | 161.48 | 161.48 | **158.11** |
| Accuracy | 67.79% | 67.99% | 66.41% | 69.20% | **71.03%** |

Table 6: Performance comparison of SPACE with different augmentation quantities on CIFAR-10-C with SNN-VGG9 model regarding **Accuracy** (%) and **Evaluation Time** (seconds) per test point.

| | No Adapt | Number of Augmentation | | | | | | |
|---|---|---|---|---|---|---|---|---|
| | | 2 | 4 | 8 | 16 | 32 | 64 | 128 |
| Accuracy | 66.57 | 54.60 | 66.91 | 69.77 | 70.60 | 71.03 | 71.07 | 71.11 |
| Evaluation Time | 0.144 | 0.318 | 0.320 | 0.327 | 0.339 | 0.345 | 0.443 | 0.788 |

emphasizing individual channel stability leads to better generalization and adaptation. These findings underscore the distinct roles of each channel in SNNs and further highlight the practical limitations of comparing entire feature maps.

**Effect of SPACE on Feature Map Similarity** further investigate the impact of SPACE adaptation on the feature map similarity, we analyzed the distribution of similarity values (computed using Equation 4) across the CIFAR-10-C dataset. As shown in Figure 3b, we observe a noticeable increase in the similarity between the original samples and their augmented counterparts after applying SPACE adaptation, particularly in the Impulse Noise domain. The similarity distributions between spike counts based feature maps in these domains indicate that the feature map consistency improves following the adaptation process. Specifically, the "After SPACE Adaptation" distributions shift towards higher similarity values, reflecting the enhanced alignment of feature maps across augmented samples. This demonstrates the effectiveness of SPACE adaptation in increasing the stability of feature representations and improving generalization under various augmentation conditions.

**Analyses of Computational Cost** To evaluate computational efficiency, we compared the GFLOPs of different baselines with the same augmented batch size, and SPACE achieved the lowest GFLOPs, demonstrating its efficiency (refers to Table 5). Regarding the additional cost of optimizing spike-behavior consistency, this step is performed only once per input in single-sample setting, without requiring whole model backpropagation or iterative updates.

**Effect of Augmentation Quantity** In the main paper, we follow MEMO [56] to determine the number of augmentations $M$. To assess $M$'s effect, we evaluated the average accuracy and the evaluation time per test point under the same computing resource conditions on CIFAR-10-C using SNN-VGG9 with varying augmentation quantities $M = \{2, 4, 8, 16, 32, 64, 128\}$. As shown in Table 6, at least 8 augmentations are required to achieve noticeable performance improvement. For $M = \{32, 64, 128\}$, while the performance continues to improve gradually, the evaluation time increases significantly. Considering the trade-off between accuracy and evaluation time, $M = 32$ is the optimal choice.

More ablation studies are presented in Appendix D.

## 5 Conclusion

In this work, we propose SPike-Aware Consistency Enhancement (SPACE), the first source-free and single-instance TTA method tailored for SNNs. By leveraging the unique spike-driven dynamics of SNNs, SPACE optimizes spike-behavior-based feature map consistency across augmented samples, directly addressing the limitations of ANN-based TTA methods that overlook the temporal and sparse nature of SNN computations. Experiments on CIFAR-10, CIFAR-100, Tiny-ImageNet, ImageNet, and DVS Gesture series distribution shifted datasets show that SPACE consistently improves performance under severe corruptions and generalizes well across architectures and backbones, demonstrating robust, practical resilience to real-world domain shifts.

## Acknowledgments and Disclosure of Funding

This work was supported in part by Chow Sang Sang Research Fund 9229161, CityU SGP grant 9380132 and ITF MSRP grant ITS/018/22MS. Any opinions, findings, conclusions, or recommendations expressed in this material do not reflect the views of the Government of the Hong Kong Special Administrative Region, the Innovation and Technology Commission, or the Innovation and Technology Fund Research Projects Assessment Panel.

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

# A    Discussion

**Selection of Alignment Layer**    We compared aligning the classifier's feature map with aligning extractor layers used in the main paper. Empirically, classifier-level alignment occasionally degrades performance on several corruptions, whereas extractor-level alignment is consistently more stable. We attribute this to: 1) classifier features being highly task-specific and high-curvature, hence small per-sample TTA steps can move decision boundaries and induce catastrophic forgetting [5]; 2) lower signal-to-noise at the classifier under pseudo-label or consistency noise, which amplifies domain-specific biases; and 3) greater adaptation overhead when updating deeper, larger parameter blocks. In contrast, channel-wise alignment at mid-level layers promotes augmentation-invariant structure while keeping the classifier intact, offering a better stability–plasticity trade-off with lower compute.

**Potential Information Loss in Equation 4**    We aggregate spikes over time, convert discrete spike counts into continuous probability values via a softmax function, then calculate the similarity. This transformation is the intentional discarding of the absolute magnitude of spike counts to focus on the relative importance of neurons within a channel. By applying softmax, we normalize away noisy fluctuations in absolute firing rates and create a stable spatial saliency map. Aligning these maps forces the model to learn which spatial features are consistently most important, directly encouraging the learning of invariant representations. In addition, while our alignment objective operates on this spatially-focused abstraction, it simultaneously preserves and refines the inherent temporal characteristics of the SNN. The underlying model remains a fully temporal processor, and the spike counts are the direct result of its complex dynamics. By demanding consistency in the outcome (the count-based saliency map), our method implicitly forces the network to stabilize the cause (the spike generation process). This creates a powerful regularization pressure that reduces temporal jitter and enhances the robustness of spike timing, all without the computational cost and overfitting risks associated with direct, spike-for-spike temporal alignment.

**Why not cosine or KL similarity**    We compute similarity as a per-channel dot product over softmax-normalized spike counts, which is bounded, symmetric, and well-defined even when raw counts are zero. Cosine similarity requires $\ell_2$ normalization and is numerically fragile under SNN sparsity: many channels yield zero or near-zero vectors, causing division-by-zero or unstable gradients; adding an $\varepsilon$ fixes numerics but introduces a sensitive hyperparameter and slightly degrades performance in our trials. KL divergence, $\mathrm{KL}(\mathbf{P}\|\mathbf{Q})$, is asymmetric and becomes ill-conditioned when entries approach zero (common with peaked softmax under corruptions), leading to exploding losses unless additional smoothing is used, which adding tuning burden again. A symmetrized variant (e.g., JSD) alleviates this but increased compute without improving over the simple dot product. Given our per-sample TTA regime, the dot product on per-channel probability maps offers the best robustness–efficiency trade-off, avoiding normalization pitfalls while retaining discriminative alignment.

**Feasibility of Deployment**    On neuromorphic platforms like SpiNNaker2 [14], the data pre-processing unit and neural network inference module are typically separate. Batch data augmentation can be performed in the pre-processing stage, outside the neuromorphic core, before feeding data into the system for inference. This design allows augmentation to be handled efficiently without impacting the neuromorphic computing core, making our method feasible on such platforms.

**Limitation**    Proposed SPACE's reliance on backpropagation may limit its applicability in resource-constrained scenarios. Future work will explore alternative optimization strategies to expand its practicality for more neuromorphic hardware and energy-efficient AI.

# B    Kernel-Based Consistency Regularization

To investigate whether higher-dimensional feature relationships can enhance the consistency measure, we integrate kernel embeddings into the loss function. The channel-wise feature probability distribution $\mathbf{P}$ can be mapped into a higher-dimensional reproducing kernel Hilbert space (RKHS) $\mathcal{H}$, using a Gaussian kernel defined as

$$k(\mathbf{z}, \mathbf{z}') = \exp(-\frac{\|\mathbf{z} - \mathbf{z}'\|^2}{2\sigma^2}), \tag{6}$$

Table 7: Performance comparison of SPACE without/with kernel embedding (*k.e.*) on CIFAR-10-C with SNN-VGG9 model regarding **Accuracy** (%). The **bold** number indicates the best result.

| Method | Noise | | | Blur | | | | | Weather | | | | Digital | | | Average |
| | Gauss. | Shot | Impl. | Defoc. | Glass | Motion | Zoom | Snow | Fog | Frost | Brit. | Contr. | Elas. | Pix. | JPEG | Acc. |
|---|---|---|---|---|---|---|---|---|---|---|---|---|---|---|---|---|
| No Adapt | 72.38 | 74.70 | 58.57 | 63.05 | 63.96 | 64.44 | 71.33 | 76.32 | 43.57 | 75.72 | 82.44 | 22.54 | 75.01 | 70.25 | 84.28 | 66.57 |
| SPACE | **77.98** | **79.34** | **69.41** | **71.59** | **67.76** | 72.14 | **74.67** | 78.43 | **52.80** | 79.59 | 83.22 | **23.85** | **75.49** | **76.24** | 82.88 | **71.03** |
| SPACE + *k.e.* | 77.85 | 79.34 | 69.25 | 71.42 | 67.64 | **72.21** | 74.61 | **78.48** | 52.46 | **79.66** | **83.25** | 23.40 | 75.42 | 76.13 | **82.98** | 70.94 |

where $\sigma$ is the kernel bandwidth. The kernel function $k(\mathbf{z}, \mathbf{z}')$ implicitly defines a mapping $\phi$: $\mathcal{Z} \to \mathcal{H}$, such that the inner product in $\mathcal{H}$ can be given by the kernel function:

$$k(\mathbf{z}, \mathbf{z}') = \langle \phi(\mathbf{z}), \phi(\mathbf{z}') \rangle_{\mathcal{H}}. \tag{7}$$

This allows us to compute relationships between feature vectors in a high-dimensional space without explicitly constructing $\phi(\cdot)$. To align feature distributions from augmented samples, we use the Mean Embedding of Distributions, which maps $\mathbf{P}_c(\mathbf{x}_i)$ into a single point $\mu_{\mathbf{P}_c(\mathbf{x}_i)}$ in RKHS. Specifically, the mean embedding is defined as

$$\mu_{\mathbf{P}_c(\mathbf{x}_i)} = \mathbb{E}_{\mathbf{z}_i \sim \mathbf{P}_c(\mathbf{x}_i)}[\phi(\mathbf{z}_i)]. \tag{8}$$

Given samples $\mathbf{z}_i \sim \mathbf{P}_c(\mathbf{x}_i)$, the embedding can be approximated as

$$\mu_{\mathbf{P}_c(\mathbf{x}_i)} \triangleq \frac{1}{D} \sum_{d=1}^{D} \phi(\mathbf{z}_{i_d}). \tag{9}$$

To measure the discrepancy between distributions $\mathbf{P}_c(\mathbf{x}_i)$ and $\mathbf{P}_c(\mathbf{x}_j)$ in RKHS, we employ the Maximum Mean Discrepancy (MMD):

$$\text{MMD}^2(\mathbf{P}_c(\mathbf{x}_i), \mathbf{P}_c(\mathbf{x}_j)) = \|\mu_{\mathbf{P}_c(\mathbf{x}_i)} - \mu_{\mathbf{P}_c(\mathbf{x}_j)}\|_{\mathcal{H}}^2. \tag{10}$$

Using the kernel trick, this can be computed without explicitly constructing $\phi(\cdot)$:

$$\text{MMD}^2(\mathbf{P}_c(\mathbf{x}_i), \mathbf{P}_c(\mathbf{x}_j)) = \frac{1}{D^2} \left\| \sum_{d=1}^{D} \phi(\mathbf{z}_{i_d}) - \sum_{d=1}^{D} \phi(\mathbf{z}_{j_d}) \right\|_{\mathcal{H}}^2. \tag{11}$$

Then we average the MMD values across channels:

$$\text{MMD}^2(i, j | \mathbf{x}) = \frac{1}{C} \sum_{c=1}^{C} \text{MMD}^2(\mathbf{P}_c(\mathbf{x}_i), \mathbf{P}_c(\mathbf{x}_j)). \tag{12}$$

Finally, we integrate the average MMD values into the original loss function $\mathcal{L}$:

$$\mathcal{L}^*(E_{\theta_E}; \mathbf{x}) \triangleq \mathcal{L} + \lambda_{\text{MMD}} \sum_{1 \leq j < i \leq M} \text{MMD}^2(i, j | \mathbf{x}) \tag{13}$$

We evaluated augmenting SPACE with a kernel embedding loss (MMD) on CIFAR-10-C using SNN-VGG9 (Table 7). The average accuracy is virtually unchanged and occasionally slightly worse than our base alignment. We hypothesize three reasons. 1) Representation sufficiency: the spike-count feature map, coupled with our linear, channel-wise similarity, already captures the essential distributional alignment across augmentations; mapping to an RKHS adds little signal. 2) Regime mismatch: our augmentations are semantics-preserving and bounded in severity (e.g., AugMix [19]), inducing small, approximately uni-modal shifts where MMD is often helpful under large or multi-modal discrepancies and provides limited additional benefit. 3) Estimation and cost: per-sample TTA affords only a small augmented batch, making MMD estimates (and their gradients) higher-variance and bandwidth-sensitive, while adding nontrivial compute. These factors combined suggest that kernel embedding is not particularly beneficial in this experimental setting.

# C  Additional Experimental Information

## C.1  Details on Datasets

In this paper, we conducted six static datasets and one neuromorphic dataset to evaluate the out-of-distribution generalization ability. They are CIFAR-10-C[2], CIFAR-100-C[3], Tiny-ImageNet-C[4] [18], ImageNet-V2[5] [42], ImageNet-R[6] [20], ImageNet-A[7] [21] and DVS Gesture-C[8] [23].

**CIFAR-10-C, CIFAR-100-C, and Tiny-ImageNet-C**  These three datasets are commonly used benchmarks for evaluating the robustness of models under various types of input corruptions. They are derived from the original CIFAR-10, CIFAR-100, and Tiny-ImageNet datasets by applying 15 corruption types (*e.g.*, noise, blur, weather, and digital distortions) at 5 severity levels to the original CIFAR-10, CIFAR-100 [31], and Tiny-ImageNet [7] datasets, respectively. CIFAR-10-C and CIFAR-100-C consist of 10 and 100 classes, each with 50,000 training images and 10,000 testing images. Tiny-ImageNet-C, derived from Tiny-ImageNet, includes 200 classes with 100,000 training images and 10,000 validation images.

**ImageNet-V2, ImageNet-R and ImageNet-A**  These three datasets are designed to evaluate model robustness under different real-world challenges. ImageNet-V2 consists of a new set of images collected under similar conditions as the original ImageNet [7], serving as a test of distribution shift. ImageNet-R focuses on image classification robustness across various artistic renditions, such as paintings, sketches, and cartoons, covering 200 ImageNet classes. ImageNet-A, on the other hand, is an adversarially filtered subset of natural images that models often misclassify, targeting their inherent weaknesses. Compared to corruption-based datasets, these benchmarks provide a better reflection of real-world scenarios and challenges.

**DVS Gesture-C**  DVS Gesture-C is a corrupted variant of the standard DVS Gesture [1] dataset. This variant introduces six corruption types: DropPixel, DropEvent, RefractoryPeriod, TimeJitter, SpatialJitter, and UniformNoise. These corruptions, implemented via the Tonic API [35], effectively simulate real-world imperfections in event-based data, including sensor noise and timing inaccuracies.

## C.2  Details on Pretrained Models

**SNN-VGG with BNTT**  [9] We adopt SNN-VGG with Batch-Normalization Through Time (BNTT) [29] as our main backbone model. BNTT decouples parameters along the time axis within each layer, effectively capturing spikes' temporal dynamics. The temporally evolving parameters in BNTT enable neurons to regulate spike rates across time steps, facilitating low-latency, low-energy training from scratch. The pre-trained model achieves accuracies of 90.61%, 69.50%, and 58.36% on the CIFAR-10, CIFAR-100, and Tiny-ImageNet test sets, respectively.

**SNN-ResNet**  [10] To explore other CNN architecture, we use SNN-ResNet11 [32], which trains SNNs directly via surrogate gradient. The pre-trained model used in our experiments can achieve 90.95% accuracy on the original CIFAR-10 test sets.

**Spike-driven Transformer V3**  [11] To evaluate the effectiveness of our work on more complex networks and datasets, we utilized a 19M parameter model in Spike-driven Transformer V3 [52], which achieves 79.60% accuracy on the ImageNet-1K validation set. This model has a patch size of $14 \times 14$, with an embedding dimension of 360 output by the final feature extractor. Instead of

---

[2]https://zenodo.org/records/2535967

[3]https://zenodo.org/records/3555552

[4]https://zenodo.org/records/2469796

[5]https://huggingface.co/datasets/vaishaal/ImageNetV2/tree/main

[6]https://github.com/hendrycks/imagenet-r

[7]https://github.com/hendrycks/natural-adv-examples

[8]http://research.ibm.com/dvsgesture/

[9]https://github.com/Intelligent-Computing-Lab-Yale/BNTT-Batch-Normalization-Through-Time

[10]https://github.com/chan8972/Enabling_Spikebased_Backpropagation

[11]https://github.com/BICLab/Spike-Driven-Transformer-V3

Table 8: Performance comparison of SPACE with different augmentation strength on CIFAR-10-C (Gaussian Noise) with SNN-VGG9 model regarding **Accuracy** (%).

| Strength $s$ | 1 | 3 | 5 | 7 | 10 | Larger Boundary |
|---|---|---|---|---|---|---|
| Accuracy | 77.98 | 77.88 | 77.48 | 77.41 | 77.02 | 74.52 |

using the class token, the entire embedding feature map is employed as the input to the classifier. This design naturally aligns with our method, where the channel dimension $C$ corresponds to the embedding dimension, and the spatial dimension $D$ matches the patch size.

**SNN-ConvLSTM**   DVS Gesture is a neuromorphic dataset with a temporal dimension, making it well-suited for evaluating RNN-based architectures. To validate the effectiveness of our work on such networks, we employed a spike-driven 2ConvLSTM-1FC model, with hidden layer dimensions of 32 and 64. This network achieves an accuracy of 93.36% on the original DVS Gesture test set.

### C.3   Details on Implementation

To better reflect real-world scenarios, we adapt our method to independently perform adaptation on each individual test input. In the same experiment, the learning rate remains fixed across different shifted domains, as the domain of a given test sample is typically unknown in real-world settings. For each set of experiments, we employ the SGD optimizer and evaluate $\eta$ ranging from 0.001 to 0.8, selecting the optimal value based on performance. We trained and tested all models and datasets in 4 NVIDIA GeForce RTX 3090 GPUs.

## D   Additional Ablation Studies

**Effect of Augmentation Strength**   We study the impact of AugMix [19] strength $s$ on SPACE. Following MEMO, our default is $s{=}1$; we sweep $s \in [1, 10]$ while keeping all other AugMix settings fixed (mixture width/depth, operators) and respecting AugMix's semantic bounds (e.g., rotation $\leq 30°$, translation $\leq \pm16$ px). As shown in Table 8, accuracy remains stable within these bounds, with only minor declines as $s$ increases. When we deliberately exceed the bounds (e.g., rotation $50°$, translation $\pm18$ px), accuracy drops noticeably, indicating that overly aggressive transformations inject harmful noise and destabilize per-sample alignment under the augmented batches. In this regime, the alignment objective tends to match augmentation-induced artifacts (noise) rather than invariant, class-relevant structure in the feature maps, thereby undermining TTA and diminishing its gains. These results are consistent with AugMix's design goal of bounded, semantics-preserving perturbations; we therefore adopt $s{=}1$ by default and recommend staying within AugMix's prescribed ranges for robust adaptation.

**Effect of Encoding Methods for SNNs**   We assess sensitivity to spike encoding by replacing Poisson rate coding with temporal coding in SNN-VGG9 while keeping architecture, training, and TTA hyperparameters fixed. As reported in Table 9, the temporal-coded baseline is markedly less robust under domain shift than its rate-coded counterpart. This aligns with intuition: temporal coding relies on precise spike times and is therefore vulnerable to perturbation-induced timing jitter, whereas rate coding averages over spikes, providing redundancy and noise tolerance.

Table 10 shows that SPACE improves both models, with consistently larger relative gains for the temporal-coded network.   The mechanism is a temporal-boundary effect: the total spike count is a sensitive proxy for timing stability because small perturbations can shift spikes across the fixed processing window; for spikes near the boundary, tiny jitters flip their contribution from 1 to 0, causing discrete

Table 9: Performance of SNNs with different encoding under domain shift. Values are Top-1 accuracy (%); Acc. Loss = Clean - Shifted.

| Encoding Method | Clean | Shifted | Acc. Loss |
|---|---|---|---|
| Temporal Coding | 89.27 | 58.11 | **31.16** |
| Rate Coding | 90.61 | 66.57 | **24.04** |

changes in the count. By enforcing consistency of total counts across augmentations, SPACE penalizes these boundary-crossing events and implicitly forces the model to stabilizes spike timing without

Table 10: Performance comparison of spike encoding methods for SNN-VGG9 model on CIFAR-10-C regarding **Accuracy** (%). The **bold** number indicates the performance improvement.

| Method | Noise | | | Blur | | | | | Weather | | | | Digital | | | Average |
|---|---|---|---|---|---|---|---|---|---|---|---|---|---|---|---|---|
| | Gauss. | Shot | Impl. | Defoc. | Glass | Motion | Zoom | Snow | Fog | Frost | Brit. | Contr. | Elas. | Pix. | JPEG | Acc. |
| *Temporal Coding* | | | | | | | | | | | | | | | | |
| No Adapt | 39.74 | 39.32 | 68.42 | 59.59 | 54.43 | 60.19 | 65.90 | 69.78 | 46.21 | 66.77 | 78.14 | 27.26 | 70.29 | 46.00 | 79.59 | 58.11 |
| SPACE | 45.19 | 44.00 | 71.82 | 64.16 | 56.31 | 64.90 | 68.65 | 70.29 | 56.90 | 72.40 | 79.09 | 68.64 | 70.66 | 50.06 | 80.07 | 64.21 (**+6.10**) |
| *Rate Coding* | | | | | | | | | | | | | | | | |
| No Adapt | 72.38 | 74.70 | 58.57 | 63.05 | 63.96 | 64.44 | 71.33 | 76.32 | 43.57 | 75.72 | 82.44 | 22.54 | 75.01 | 70.25 | 84.28 | 66.57 |
| SPACE | 77.98 | 79.34 | 69.41 | 71.59 | 67.76 | 72.14 | 74.67 | 78.43 | 52.80 | 79.59 | 83.22 | 23.85 | 75.49 | 76.24 | 82.88 | 71.03 (**+4.46**) |

Table 11: Performance comparison of SPACE with different objectives on CIFAR-10-C with SNN-VGG9 model regarding **Accuracy** (%). The **bold** number indicates the best result. Average membrane potential and spike through all time steps are referred as *a.m.p.* and *s.t.t.* respectively.

| Method | Noise | | | Blur | | | | | Weather | | | | Digital | | | Average |
|---|---|---|---|---|---|---|---|---|---|---|---|---|---|---|---|---|
| | Gauss. | Shot | Impl. | Defoc. | Glass | Motion | Zoom | Snow | Fog | Frost | Brit. | Contr. | Elas. | Pix. | JPEG | Acc. |
| No Adapt | 72.38 | 74.70 | 58.57 | 63.05 | 63.96 | 64.44 | 71.33 | 76.32 | 43.57 | 75.72 | 82.44 | 22.54 | 75.01 | 70.25 | 84.28 | 66.57 |
| SPACE | **77.98** | **79.34** | **69.41** | **71.59** | **67.76** | **72.14** | **74.67** | **78.43** | **52.80** | **79.59** | **83.22** | **23.85** | 75.49 | **76.24** | 82.88 | **71.03** |
| SPACE + *a.m.p.* | 74.87 | 76.05 | 61.64 | 64.90 | 65.79 | 65.88 | 72.22 | 77.66 | 45.66 | 76.64 | 82.81 | 21.57 | **75.53** | 70.91 | **84.74** | 67.79 |
| SPACE + *s.t.t.* | 72.74 | 73.84 | 57.86 | 62.91 | 63.66 | 64.33 | 70.94 | 76.35 | 43.16 | 74.77 | 82.30 | 21.20 | 74.24 | 69.35 | 83.97 | 66.11 |

brittle spike-to-spike matching. The resulting gradients are stronger in temporal-coded models than in rate-coded ones, yielding larger robustness gains.

**Exploration of Alternative Alignment Objectives**   We evaluate two alternatives to SPACE's default count-based feature map on CIFAR-10-C with SNN-VGG9, keeping architecture, training, and TTA hyperparameters fixed. Results are summarized in Table 11.

1) Average Membrane Potential (SPACE + *a.m.p.*). Let $\mathbf{U}(\mathbf{x}) \in \mathbb{R}^{T \times C \times D}$ denote the LIF membrane traces from the selected layer (see subsection 3.2). For spikes, the post-reset potential is used. We form $\mathbf{F}(\mathbf{x}) = \frac{1}{T} \sum_{t=1}^{T} \mathbf{U}_t(\mathbf{x}) \in \mathbb{R}^{C \times D}$ and apply the same channel-wise consistency loss as SPACE. It improves over No Adapt but is consistently worse than count-based SPACE on most corruptions. Membrane traces are smooth and highly correlated across augmentations, making them less sensitive to discrete spiking changes; near-threshold, non-firing trajectories can be indistinguishable in $\mathbf{F}(\mathbf{x})$, whereas counts capture such events directly.

2) Spikes Through Time (SPACE + *s.t.t.*). Instead of summing over time, we retain the temporal dimension of the spike tensor $\mathbf{S}(\mathbf{x}) \in \mathbb{R}^{T \times C \times D}$ and align per-channel activity at each time step across augmentations (loss averaged over $t$); optimization is unchanged. This variant does not improve robustness. Keeping the full temporal waveform inflates variance and forces the model to match augmentation-specific transients: benign timing jitter (e.g., a spike at $t=10$ vs. $t=12$) is penalized as a mismatch, producing conflicting gradients that push spike times to align across views rather than stabilizing the underlying excitability. The model thus overfits to view-specific temporal shapes, hurting generalization, while also incurring higher compute and memory. We further applied Gaussian smoothing along time before alignment, $\tilde{\mathbf{S}} = g_\sigma * \mathbf{S}$, to reduce jitter-induced overfitting. Results remained suboptimal. With small $\sigma$, smoothing is too weak to suppress shift-induced conflicts; with larger $\sigma$, it erases informative peaks and attenuates counts. The core issue is not merely information loss but the learning signal each representation provides: spike counts yield a shift-invariant, macro-level target that is robust to timing jitter and directs clear adjustments to neuron excitability, whereas smoothed sequences still demand matching temporal shapes, preserving shift-sensitivity and inducing noisy, contradictory gradients.

Consequently, SPACE's count-based objective offers a more stable and effective alignment signal for our TTA setting.

**Statistical Significance**   We assess statistical significance using the Wilcoxon signed-rank test. Specifically, for each of the three main experiments, we run five independent trials and apply a paired Wilcoxon test between SPACE and each baseline across the five runs; reported $p$-values in Table 12 confirm that SPACE's gains are statistically significant. To further strengthen the evidence, we select Gaussian and Shot Noise from CIFAR-10-C and conduct Wilcoxon signed-rank tests at the per-class level (Table 13), demonstrating that the improvements are consistent across categories rather than driven by a few classes.

Table 12: Wilcoxon signed-rank test on CIFAR-10-C, CIFAR-100-C, and Tiny-ImageNet-C with CNN-based SNN models. W denotes the Wilcoxon signed-rank statistic and p is the two-sided p-value.

| Compared Method | Metric | Noise | | | Blur | | | | | Weather | | | | Digital | | | Average |
|---|---|---|---|---|---|---|---|---|---|---|---|---|---|---|---|---|---|
| | | Gauss. | Shot | Impl. | Defoc. | Glass | Motion | Zoom | Snow | Fog | Frost | Brit. | Contr. | Elas. | Pix. | JPEG | Acc. |
| *CIFAR-10-C, SNN-VGG9* | | | | | | | | | | | | | | | | | |
| vs. No Adapt | W | 15 | 15 | 15 | 15 | 15 | 15 | 15 | 15 | 15 | 15 | 15 | 15 | 15 | 15 | 0 | 15 |
| | p | 0.0312 | 0.0312 | 0.0312 | 0.0312 | 0.0312 | 0.0312 | 0.0312 | 0.0312 | 0.0312 | 0.0312 | 0.0312 | 0.0312 | 0.0312 | 0.0312 | 1.0000 | 0.0312 |
| vs. RoTTA | W | 15 | 15 | 15 | 15 | 15 | 15 | 15 | 15 | 15 | 15 | 15 | 15 | 15 | 15 | 0 | 15 |
| | p | 0.0312 | 0.0312 | 0.0312 | 0.0312 | 0.0312 | 0.0312 | 0.0312 | 0.0312 | 0.0312 | 0.0312 | 0.0312 | 0.0312 | 0.0312 | 0.0312 | 1.0000 | 0.0312 |
| vs. DeYO | W | 15 | 15 | 15 | 15 | 15 | 15 | 15 | 15 | 15 | 15 | 15 | 15 | 15 | 15 | 15 | 15 |
| | p | 0.0312 | 0.0312 | 0.0312 | 0.0312 | 0.0312 | 0.0312 | 0.0312 | 0.0312 | 0.0312 | 0.0312 | 0.0312 | 0.0312 | 0.0312 | 0.0312 | 0.0312 | 0.0312 |
| vs. SITA | W | 15 | 15 | 15 | 15 | 15 | 15 | 15 | 15 | 15 | 15 | 15 | 15 | 15 | 15 | 0 | 15 |
| | p | 0.0312 | 0.0312 | 0.0312 | 0.0312 | 0.0312 | 0.0312 | 0.0312 | 0.0312 | 0.0312 | 0.0312 | 0.0312 | 0.0312 | 0.0312 | 0.0312 | 1.0000 | 0.0312 |
| vs. MEMO | W | 15 | 0 | 15 | 15 | 15 | 15 | 15 | 15 | 15 | 15 | 10 | 9 | 0 | 15 | 0 | 15 |
| | p | 1.0000 | 0.0312 | 0.0312 | 0.0312 | 0.0312 | 0.0312 | 0.0312 | 0.0312 | 0.0312 | 0.0312 | 0.4000 | 0.3333 | 0.0312 | 0.0312 | 1.0000 | 0.0312 |
| *CIFAR-100-C, SNN-VGG9* | | | | | | | | | | | | | | | | | |
| vs. No Adapt | W | 15 | 15 | 15 | 15 | 15 | 15 | 15 | 15 | 15 | 15 | 15 | 15 | 15 | 15 | 15 | 15 |
| | p | 0.0312 | 0.0312 | 0.0312 | 0.0312 | 0.0312 | 0.0312 | 0.0312 | 0.0312 | 0.0312 | 0.0312 | 0.0312 | 0.0312 | 0.0312 | 0.0312 | 0.0312 | 0.0312 |
| vs. RoTTA | W | 15 | 15 | 15 | 15 | 15 | 15 | 15 | 15 | 15 | 15 | 15 | 15 | 15 | 15 | 15 | 15 |
| | p | 0.0312 | 0.0312 | 0.0312 | 0.0312 | 0.0312 | 0.0312 | 0.0312 | 0.0312 | 0.0312 | 0.0312 | 0.0312 | 0.0312 | 0.0312 | 0.0312 | 0.0312 | 0.0312 |
| vs. DeYO | W | 15 | 15 | 15 | 15 | 15 | 15 | 15 | 15 | 15 | 15 | 15 | 15 | 15 | 15 | 15 | 15 |
| | p | 0.0312 | 0.0312 | 0.0312 | 0.0312 | 0.0312 | 0.0312 | 0.0312 | 0.0312 | 0.0312 | 0.0312 | 0.0312 | 0.0312 | 0.0312 | 0.0312 | 0.0312 | 0.0312 |
| vs. SITA | W | 15 | 15 | 15 | 15 | 15 | 15 | 15 | 15 | 15 | 15 | 15 | 15 | 15 | 15 | 15 | 15 |
| | p | 0.0312 | 0.0312 | 0.0312 | 0.0312 | 0.0312 | 0.0312 | 0.0312 | 0.0312 | 0.0312 | 0.0312 | 0.0312 | 0.0312 | 0.0312 | 0.0312 | 0.0312 | 0.0312 |
| vs. MEMO | W | 15 | 15 | 15 | 15 | 15 | 15 | 15 | 15 | 15 | 15 | 15 | 15 | 15 | 15 | 10 | 15 |
| | p | 0.0312 | 0.0312 | 0.0312 | 0.0312 | 0.0312 | 0.0312 | 0.0312 | 0.0312 | 0.0312 | 0.0312 | 0.0312 | 0.0312 | 0.0312 | 0.0312 | 0.4000 | 0.0312 |
| *Tiny-ImageNet-C, SNN-VGG11* | | | | | | | | | | | | | | | | | |
| vs. No Adapt | W | 15 | 15 | 15 | 15 | 15 | 15 | 15 | 15 | 15 | 15 | 15 | 15 | 15 | 15 | 15 | 15 |
| | p | 0.0312 | 0.0312 | 0.0312 | 0.0312 | 0.0312 | 0.0312 | 0.0312 | 0.0312 | 0.0312 | 0.0312 | 0.0312 | 0.0312 | 0.0312 | 0.0312 | 0.0312 | 0.0312 |
| vs. RoTTA | W | 15 | 15 | 15 | 15 | 15 | 15 | 15 | 15 | 15 | 15 | 15 | 15 | 15 | 15 | 15 | 15 |
| | p | 0.0312 | 0.0312 | 0.0312 | 0.0312 | 0.0312 | 0.0312 | 0.0312 | 0.0312 | 0.0312 | 0.0312 | 0.0312 | 0.0312 | 0.0312 | 0.0312 | 0.0312 | 0.0312 |
| vs. DeYO | W | 15 | 15 | 15 | 15 | 15 | 15 | 15 | 15 | 15 | 15 | 15 | 15 | 15 | 15 | 15 | 15 |
| | p | 0.0312 | 0.0312 | 0.0312 | 0.0312 | 0.0312 | 0.0312 | 0.0312 | 0.0312 | 0.0312 | 0.0312 | 0.0312 | 0.0312 | 0.0312 | 0.0312 | 0.0312 | 0.0312 |
| vs. SITA | W | 15 | 15 | 15 | 15 | 15 | 15 | 15 | 15 | 15 | 15 | 15 | 15 | 15 | 15 | 15 | 15 |
| | p | 0.0312 | 0.0312 | 0.0312 | 0.0312 | 0.0312 | 0.0312 | 0.0312 | 0.0312 | 0.0312 | 0.0312 | 0.0312 | 0.0312 | 0.0312 | 0.0312 | 0.0312 | 0.0312 |
| vs. MEMO | W | 15 | 15 | 15 | 15 | 15 | 15 | 15 | 15 | 15 | 15 | 15 | 15 | 15 | 15 | 15 | 15 |
| | p | 0.0312 | 0.0312 | 0.0312 | 0.0312 | 0.0312 | 0.0312 | 0.0312 | 0.0312 | 0.0312 | 0.0312 | 0.0312 | 0.0312 | 0.0312 | 0.0312 | 0.0312 | 0.0312 |

Table 13: Wilcoxon signed-rank test at the per-class level on CIFAR-10-C (Gaussian/Shot Noise) with CNN-based SNN models. W denotes the Wilcoxon signed-rank statistic and p is the two-sided p-value.

| Compared Method | Metric | Class Index | | | | | | | | | |
|---|---|---|---|---|---|---|---|---|---|---|---|
| | | 0 | 1 | 2 | 3 | 4 | 5 | 6 | 7 | 8 | 9 |
| *Gaussian Noise* | | | | | | | | | | | |
| vs. No Adapt | W | 15 | 15 | 15 | 15 | 15 | 15 | 15 | 15 | 15 | 15 |
| | p | 0.0312 | 0.0312 | 0.0312 | 0.0312 | 0.0312 | 0.0312 | 0.0312 | 0.0312 | 0.0312 | 0.0312 |
| vs. RoTTA | W | 15 | 15 | 15 | 15 | 15 | 15 | 15 | 15 | 15 | 15 |
| | p | 0.0312 | 0.0312 | 0.0312 | 0.0312 | 0.0312 | 0.0312 | 0.0312 | 0.0312 | 0.0312 | 0.0312 |
| vs. DeYO | W | 15 | 15 | 15 | 15 | 15 | 15 | 15 | 15 | 15 | 15 |
| | p | 0.0312 | 0.0312 | 0.0312 | 0.0312 | 0.0312 | 0.0312 | 0.0312 | 0.0312 | 0.0312 | 0.0312 |
| vs. SITA | W | 15 | 15 | 15 | 15 | 15 | 15 | 15 | 15 | 15 | 15 |
| | p | 0.0312 | 0.0312 | 0.0312 | 0.0312 | 0.0312 | 0.0312 | 0.0312 | 0.0312 | 0.0312 | 0.0312 |
| vs. MEMO | W | 15 | 15 | 15 | 15 | 15 | 15 | 15 | 15 | 15 | 15 |
| | p | 0.0312 | 0.0312 | 0.0312 | 0.0312 | 0.0312 | 0.0312 | 0.0312 | 0.0312 | 0.0312 | 0.0312 |
| *Shot Noise* | | | | | | | | | | | |
| vs. No Adapt | W | 15 | 15 | 15 | 15 | 15 | 15 | 15 | 15 | 15 | 15 |
| | p | 0.0312 | 0.0312 | 0.0312 | 0.0312 | 0.0312 | 0.0312 | 0.0312 | 0.0312 | 0.0312 | 0.0312 |
| vs. RoTTA | W | 15 | 15 | 15 | 15 | 15 | 15 | 15 | 15 | 15 | 15 |
| | p | 0.0312 | 0.0312 | 0.0312 | 0.0312 | 0.0312 | 0.0312 | 0.0312 | 0.0312 | 0.0312 | 0.0312 |
| vs. DeYO | W | 15 | 15 | 15 | 15 | 15 | 15 | 15 | 15 | 15 | 15 |
| | p | 0.0312 | 0.0312 | 0.0312 | 0.0312 | 0.0312 | 0.0312 | 0.0312 | 0.0312 | 0.0312 | 0.0312 |
| vs. SITA | W | 15 | 15 | 15 | 15 | 15 | 15 | 15 | 14 | 15 | 15 |
| | p | 0.0312 | 0.0312 | 0.0312 | 0.0312 | 0.0312 | 0.0312 | 0.0312 | 0.0625 | 0.0312 | 0.0312 |
| vs. MEMO | W | 5 | 0 | 0 | 5 | 5 | 0 | 0 | 0 | 14 | 0 |
| | p | 0.7812 | 1.0000 | 1.0000 | 0.7812 | 0.7812 | 1.0000 | 1.0000 | 1.0000 | 0.0625 | 1.0000 |

