# OpenReview forum: "SPACE: SPike-Aware Consistency Enhancement for Test-Time Adaptation in Spiking Neural Networks"
_NeurIPS.cc/2025/Conference — NeurIPS 2025 poster_

### Official Review · Reviewer_4hgi · 2025-06-29

**Clarity:** 3
**Significance:** 2
**Originality:** 3
**Rating:** 4
**Confidence:** 3

**Summary:**

This paper proposes a source-free single-instance test-time adaptation framework for SNNs to address the problem of their high sensitivity to distribution shifts, characteristics of real-world scenarios. Traditional test-time adaptation methods in ANNs, such as MEMO and SITA, cannot match the internal dynamics or external architecture of SNNs. To this end, the authors introduce a spike-aware consistency enhancement method, in which a spike feature map extractor is trained to mitigate distribution shifts of the augmented batch from a single instance and is then incorporated into the prediction model. Experiments demonstrate the consistent performance improvements on different network structures and datasets.

**Questions:**

1. Why didn't the TTL method consider cosine similarity when designing the similarity function?

2. Line 273: What is the number of inference steps used in Spike-driven V3 and SNN-ConvLSTM?

3. Line 15 in Appendix: What is the kernel function \phi ?

4. How is it possible to improve the similarity of domain-shifted instances during the training of the extractor while maintaining classification performance? This question is purely out of my curiosity.

5. minors:
(1) line 56: names->named
(2) line 211: F(x_1), F(x_1)->F(x_1), F(x_2)

**Ethical Concerns:**

["NO or VERY MINOR ethics concerns only"]

**Final Justification:**

I think Test-Time Adaptation is helpful for the SNN field. After reviewing the authors’ response, I raised my score.

**Limitations:**

I agree that alternative optimization strategies to expand the practicality can be explored for neuromorphic hardware and energy-efficient AI.

**Quality:**

2

**Strengths And Weaknesses:**

Strengths:

1.The observed performance improvements on various datasets and network frameworks confirm the applicability of the proposed approach.

2. The ablation study of the feature map, along with the extended investigation of kernel techniques and objectives in the appendix, is detailed and valuable.

3. The paper is well-organized and is clearly articulated.

Weaknesses:

1. Since the paper states in line 34 that“SNNs are particularly vulnerable to domain shifts,” the authors are implying that SNNs are more prone to domain‑shift issues than ANNs. Yet, they neither cite any prior work analyzing robustness or domain-shift problems of SNNs, nor present any empirical or theoretical observations to support it, nor compare SNNs to ANNs in terms of domain shift. This significantly weakens the persuasiveness of their motivation.

2. Given that this is the first work integrating SNNs and TTA, it would be valuable to compare its performance against existing ANN baselines. The authors should consider adding such experiments.

3. The description of the SPACE framework is not clear enough. In Figure 2, the SPACE framework consists of two parts: the SNN Feature Extractor and the Classifier. Could the authors please clarify which part of the neural network the Extractor refers to? According to my understanding, if a neural network has L layers, the classifier is usually the last one or two layers, so the Extractor seems to be the main body of the network. How would the performance of SPACE change if different depths of the Extractor were used (for example, dividing the network into more layers for the classifier)?

4. In Line 334, the authors mention the "global feature map" for the first time but do not explain what it means. Moreover, what is the difference between the global feature map and the local feature map?

---

> ### Author Rebuttal · Authors · 2025-07-30
>
> Thank you for your thoughtful comments, which have helped us make our work clearer and more solid. We have carefully addressed all your concerns in detail.
>
> **[W1] SNN's Vulnerability to Domain Shift.** As mentioned in [1,2,3], prior works have highlighted the robustness challenges and vulnerability of SNNs to domain shift. To further support this claim, we conducted additional experiments where we evaluated the performance of both SNNs and ANNs with the same architecture and scale on shifted datasets. The results, shown in the table below, indicate that SNNs experience a significantly larger accuracy drop compared to ANNs under domain shifts. These findings validate our motivation and emphasize the need for developing robust TTA methods for SNNs.
> |Architecture|Dataset Series|Clean Data|Shifted Data|Acc. Loss|Clean Data|Shifted Data|Acc. Loss|
> |-|-|-|-|-|-|-|-|
> ||||ANN|||SNN||
> |VGG9|CIFAR10|90.53%|77.25%|**13.28%**|90.61%|67.86%|**22.74%**|
> |VGG11|CIFAR100|70.86%|48.32%|**22.54%**|69.50%|41.35%|**28.14%**|
> |VGG11|TinyImageNet|61.24%|19.76%|**41.48%**|58.36%|14.56%|**43.80%**|
> |Transformer|ImageNet|85.16%|75.49%|**9.67%**|79.60%|67.64%|**11.96%**|
>
> **[W2] Additional Baselines.** We would like to clarify the setting of our work. In our **single-sample TTA** scenario, each sample undergoes test-time adaptation independently. This setting reflects real-world scenarios where it is often impractical to test with a mini-batch of data. Consequently, many batch-based TTA methods that rely on adjusting normalization layers [4,5] are not suitable for this setting. Moreover, some SNN architectures lack normalization layers altogether. Therefore, we initially chose SITA and MEMO, two single-instance TTA methods, as baselines. \
> For the ANN-based TTA methods such as SHOT[6] and TAST[7] mentioned by Reviewer sQFP, they are incompatible with SNNs for the following reasons:
> 1. SHOT relies on pseudo-labeling in a static feature vector space, which disregards the critical temporal dynamics of SNNs. This results in incomplete feature representation and invalidates distance-based metrics.
> 2. SHOT freezes the static classifier layer, but SNN classification is tightly coupled with spiking dynamics over time. There is no standalone, static classification layer that can be frozen without disrupting the functionality.
> 3. TAST depends on prior probabilities, which are similarly unsuitable given SNNs' temporal and discrete spiking nature.
>
> To address the your suggestion, we additionally included RoTTA[8] and the recent DeYO[9] as baselines. Since both methods require batch data, we applied the same augmentation strategy as SPACE to create a pseudo-batch for each sample to ensure a fair comparison. The results are shown below. RoTTA's memory bank cannot be effectively utilized in this single-sample setting, leading to limited performance gains. DeYO's reliance on smooth probability outputs for confidence comparison does not align with SNN's discrete, spike-based outputs, introducing significant errors. In addition, all the baselines exhibit higher computational costs (GFLOPs). Our SPACE method, designed specifically for SNNs, outperforms these baselines while maintaining lower computational cost.
> |Method||Noise|||Blur||||Weather||||Digital|||Acc.|GFLOPs
> |-|-|-|-|---------|--------|--------|--------|---------|--------|--------|------------|----------|-------------------|----------|------------------|--------|--------|
> || **Gauss.** | **Shot** | **Impl.** | **Defoc.** | **Glass** | **Motion** | **Zoom** |  **Snow**  | **Fog** | **Frost** | **Brit.** | **Contr.** | **Elas.** | **Pix.** | **JPEG** | **Avg.** ||
> |No&nbsp;Adapt|72.38%|74.70%|58.57%|63.05%|63.96%|64.44%|71.33%|76.32%|43.57%|75.72%|82.44%|22.54%|75.01%|70.25%|84.28%|66.57%|-|
> | RoTTA|73.60%|75.49%|61.61%|66.08%|67.53%|67.75%|72.40%|74.34%|51.54%|76.35%|80.98% |23.01%|74.23%|68.60%|83.29%|67.79%|160.67|
> | DeYO |73.97%|76.16%|62.45%|66.29%|67.37%|67.60%|73.52%|74.90%|50.68%|76.23%|81.30% |20.90%|75.19%|70.44%|82.85%|67.99%|318.07|
> | SITA |73.06%|74.15%|58.41%|62.94%|64.21%|64.36%|70.72%|76.67%|43.40%|75.24%|82.51% |22.02%|74.74%|69.62%|84.16%|66.41%|161.48|
> | MEMO |77.73%|79.50%|65.74%|65.61%|67.50%|66.24%|72.61%|77.38%|45.47%|78.64%|83.05% |23.83%|76.45%|73.25%|85.05%|69.20%|162.48|
> | SPACE|77.98%|79.34%|69.41%|71.59%|67.76%|72.14%|74.67%|78.43%|52.80%|79.59%|83.22% |23.85%|75.49%|76.24%|82.88%|71.03%|158.11|
>
> **[W3] More Description of Our Framework.** In our framework, the feature extractor refers to the main body of the neural network, which could be a CNN, Transformer, or ConvLSTM, excluding the last one or two fully connected layers that form the classifier. We observe the feature map of the last layer of the feature extractor. If this layer’s spatial dimension is 1×1, alignment targets become too scarce, potentially leading to overfitting. In such cases, we use the penultimate layer of the feature extractor to ensure that the alignment targets retain rich, high-level semantic information that is more compact and representative, reducing redundancy while focusing on features critical for classification. We also experimented with aligning the classifier’s feature map before, but this approach sometimes degraded performance in certain domains. We attribute this to the classifier being more sensitive to changes, which we believe makes the TTA process less stable and potentially more prone to catastrophic forgetting [10]. Moreover, aligning deeper feature maps introduces additional computational costs.
>
> **[W4] Explanation of Global Feature Map.** The "global feature map" refers to treating the entire feature map of size $C \times H \times W$ as a single entity and measuring similarity on this full representation. In contrast, the "local feature map" considers each spatial position in the $H \times W$ dimension independently, calculating similarity at each  $H \times W$ dimension and averaging across the $C$ channels. This distinction reflects whether similarity is computed globally or spatially at a finer granularity. We’ll clarify this in the revised version.
>
> **[Q1] Similarity Function.** Our method uses softmax followed by inner product to evaluate similarity, which, like cosine similarity, is based on inner product. However, cosine similarity involves dividing by the L2 norm, which can lead to division-by-zero errors due to the sparsity of SNNs, where some neurons in certain channels may produce zero spikes across all timesteps. To avoid this issue and ensure robust similarity computation, we opted for our current design.
>
> **[Q2] Inference Steps.** We follow the setting in Spike-driven Transformer V3 work [11], where the time step is set to **4**. The time steps used in ConvLSTM is **32**.
>
> **[Q3] Kernel Function $\phi$.**  $\phi$ represents an **implicit** mapping function that projects input data from the original space $\mathcal{Z}$ to a higher-dimensional RKHS $\mathcal{H}$. This mapping is implicitly defined by the kernel function $k(\mathbf{z},\mathbf{z}')$, such that the inner product in $\mathcal{H}$ is given by $\langle \phi(\mathbf{z}),\phi(\mathbf{z}')\rangle_\mathcal{H}=k(\mathbf{z},\mathbf{z}')$. Hence, $\phi$ does not need to be explicitly constructed, as all computations in $\mathcal{H}$ can be performed directly through the kernel function, which is Gaussian kernel here.
>
> **[Q4] Transfer Our Strategy on Training Phase.** This is indeed a great idea. By incorporating our strategy into the loss function during training, we can further amplify the impact of augmentation in the training phase. Due to time constraints, we implemented this approach on the ConvLSTM backbone, and the test accuracy improved from 90.23% to 91.97%. In future work, we plan to explore more efficient methods along these lines to help further narrow the gap between SNNs and ANNs.
>
> **[Q5] Typographical Errors.** Thank you for pointing that out! The errors have been corrected in the revised version.
>
> Reference \
> [1] Karilanova, Sanja, et al. "Zero-shot temporal resolution domain adaptation for spiking neural networks." arXiv preprint arXiv:2411.04760 (2024). \
> [2] Huang, Yifan, et al. "Flexible and scalable deep dendritic spiking neural networks with multiple nonlinear branching." arXiv preprint arXiv:2412.06355 (2024). \
> [3] Mei, Zaidao, Mark Barnell, and Qinru Qiu. "Unsupervised Adaptation of Spiking Networks in a Gradual Changing Environment." 2022 IEEE High Performance Extreme Computing Conference (HPEC). IEEE, 2022. \
> [4] Boudiaf, Malik, et al. "Parameter-free online test-time adaptation." Proceedings of the IEEE/CVF Conference on Computer Vision and Pattern Recognition. 2022. \
> [5] Lim, Hyesu, et al. "TTN: A Domain-Shift Aware Batch Normalization in Test-Time Adaptation." The Eleventh International Conference on Learning Representations, ICLR 2023, 2023. \
> [6] Liang, Jian, Dapeng Hu, and Jiashi Feng. "Do we really need to access the source data? source hypothesis transfer for unsupervised domain adaptation." International conference on machine learning. PMLR, 2020. \
> [7] Jang, Minguk, Sae-Young Chung, and Hye Won Chung. "Test-Time Adaptation via Self-Training with Nearest Neighbor Information." 11th International Conference on Learning Representations, ICLR 2023, 2023. \
> [8] Yuan, Longhui, Binhui Xie, and Shuang Li. "Robust test-time adaptation in dynamic scenarios." Proceedings of the IEEE/CVF Conference on Computer Vision and Pattern Recognition. 2023. \
> [9] Lee, Jonghyun, et al. "ENTROPY IS NOT ENOUGH FOR TEST-TIME ADAPTATION: FROM THE PERSPECTIVE OF DISENTANGLED FACTORS." 12th International Conference on Learning Representations, ICLR 2024. 2024. \
> [10] Chen, Kecheng, et al. "Test-time Adaptation for Foundation Medical Segmentation Model without Parametric Updates." International Conference on Computer Vision, 2025.
> [11] Lee, Chankyu, et al. "Enabling spike-based backpropagation for training deep neural network architectures." Frontiers in neuroscience 14 (2020): 497482.

---

> ### Author Response · Authors · 2025-08-03
> **Thank You and Follow Up**
>
> Dear Reviewer 4hgi,
>
> Thank you for taking the time to review our responses. Please let us know if you have any further questions or concerns. We would be happy to address them.
>
> Once again,  thanks for your time and consideration.

---

> > ### Comment · Reviewer_hnfy · 2025-08-03
> >
> > Thank you for the response. After reading it and considering other reviews, my concern about the limited applicability remains, so I will keep my original score.

---

> > > ### Author Response · Authors · 2025-08-03
> > > **Thank You and Address Your Concern**
> > >
> > > Dear Reviewer hnfy,
> > >
> > > Thank you for your time and for the discussion. We are genuinely grateful for your insightful comments, especially those that led to new experiments and have made our work more solid.
> > >
> > > Regarding the central point on motivation, which we addressed in [W5] of our response to you, we respectfully maintain our conviction that as SNNs continue to achieve performance parity, ensuring their practical robustness via methods like TTA is the critical next step for real-world deployment. We believe our work is a contribution to this important frontier.
> > >
> > > We thank you again for your time and thoughtful feedback.
> > >
> > > Sincerely, \
> > > The Authors

---

> > ### Comment · Reviewer_4hgi · 2025-08-04
> >
> > Thanks for the author's response. Most of my questions have been addressed. However, the incompatibility between current state-of-the-art ANN methods and SNNs should not be a justification for not including an ANN baseline. This is especially important given that this work is the first to apply SNNs in the TTA domain, with the motivation presented in the introduction being somewhat vague. Without ANN baselines, the paper comes across as an exercise in isolationism — celebrating SNNs in a vacuum, without engaging with the broader context of ANN research. The reader is unable to learn from the experiments the differences between using SNN and ANN in the TTA domain, as well as the advantages of using SNN for TTA. In this regard, Reviewer hnfy said what I want to say. I keep my original score and hope the authors can improve the motivation and experiment in the future.

---

> > > ### Author Response · Authors · 2025-08-06
> > > **Thank You and Looking Forward to Your Follow-up Feedback**
> > >
> > > Dear Reviewer 4hgi,
> > >
> > > Thank you sincerely for your detailed follow-up and for pushing us to strengthen the paper's core message. Your feedback that the paper must engage with the broader context of ANN research is a crucial point, and we have taken it very seriously.
> > >
> > > We apologize if our previous reply was not clear enough, as it seems there may be a misunderstanding. We want to respectfully clarify that **we did, in fact, implement and include additional ANN baselines in our rebuttal**, and we are committed to adding this entire section to our revised manuscript.
> > >
> > > Specifically, in response to your initial review, we:
> > > 1. **Integrated two prominent ANN TTA methods, RoTTA and DeYO**, into our experiments.
> > > 2. We presented a comprehensive table (included again below for your convenience) showing that our SNN-native method, SPACE, outperforms all the baselines while being more computationally efficient (lower GFLOPs).
> > >
> > > |Method||Noise|||Blur||||Weather||||Digital|||Acc.|GFLOPs
> > > |----------|----------|--------|---------|---------|--------|--------|--------|---------|--------|--------|------------|----------|-------------------|----------|------------------|--------|--------|
> > > || **Gauss.** | **Shot** | **Impl.** | **Defoc.** | **Glass** | **Motion** | **Zoom** |  **Snow**  | **Fog** | **Frost** | **Brit.** | **Contr.** | **Elas.** | **Pix.** | **JPEG** | **Avg.** ||
> > > |No&nbsp;Adapt|72.38%|74.70%|58.57%|63.05%|63.96%|64.44%|71.33%|76.32%|43.57%|75.72%|82.44%|22.54%|75.01%|70.25%|84.28%|66.57%|-|
> > > | RoTTA|73.60%|75.49%|61.61%|66.08%|67.53%|67.75%|72.40%|74.34%|51.54%|76.35%|80.98% |23.01%|74.23%|68.60%|83.29%|67.79%|160.67|
> > > | DeYO |73.97%|76.16%|62.45%|66.29%|67.37%|67.60%|73.52%|74.90%|50.68%|76.23%|81.30% |20.90%|75.19%|70.44%|82.85%|67.99%|318.07|
> > > | SITA |73.06%|74.15%|58.41%|62.94%|64.21%|64.36%|70.72%|76.67%|43.40%|75.24%|82.51% |22.02%|74.74%|69.62%|84.16%|66.41%|161.48|
> > > | MEMO |77.73%|79.50%|65.74%|65.61%|67.50%|66.24%|72.61%|77.38%|45.47%|78.64%|83.05% |23.83%|76.45%|73.25%|85.05%|69.20%|162.48|
> > > |**SPACE**|77.98%|79.34%|69.41%|71.59%|67.76%|72.14%|74.67%|78.43%|52.80%|79.59%|83.22% |23.85%|75.49%|76.24%|82.88%|**71.03%**|**158.11**|
> > >
> > > Most importantly, your feedback helped us realize a deeper point that we will now make central to our paper's narrative. Your question about what a reader learns is exactly right. Our experiments provide a critical insight:
> > >
> > > The incompatibility is **not an excuse, but a key finding**. The underperformance of powerful ANN baselines like RoTTA and DeYO in our setting reveals that directly porting ANN-centric TTA concepts (like reliance on smooth probability outputs or static feature banks) to the temporal, discrete, and event-driven paradigm of SNNs is fundamentally suboptimal.
> > >
> > > This is precisely the advantage and motivation for our work. It demonstrates that to unlock the potential of TTA for SNNs (e.g., higher efficiency), a co-designed, SNN-native strategy like SPACE is not just beneficial, but **necessary**.
> > >
> > > Based on your invaluable guidance, we will revise the introduction and discussion to explicitly frame our work this way—not as an "exercise in isolationism," but as a pioneering study that bridges the gap, revealing both the challenges and the unique opportunities of SNNs in the TTA domain.
> > >
> > > We are grateful for your time and rigorous review, which has significantly helped us improve the paper's impact. We hope this clarification addresses your primary concern, and we would be very grateful if you would consider our extensive additions and the new insights they provide.
> > >
> > > Sincerely,
> > >
> > > Authors

---

> > > > ### Author Response · Authors · 2025-08-08
> > > > **Follow-up on Our Supplementary Rebuttal**
> > > >
> > > > Dear Reviewer 4hgi,
> > > >
> > > > We are writing to gently follow up on the detailed response we sent earlier regarding your feedback.
> > > >
> > > > We just wanted to ensure you had a chance to see it and were wondering if our response and the experiments have addressed your primary concerns. We would be very grateful to hear your thoughts.
> > > >
> > > > Thank you again for your time and invaluable guidance.
> > > >
> > > > Sincerely,
> > > >
> > > > The Authors

---

> ### Author Response · Authors · 2025-08-05
> **Thank You and Address Your Concern**
>
> Dear Reviewer 4hgi,
>
> Thanks for your reply. We have discussed and included additional ANN baselines in our rebuttal (refers to **[W2] Additional Baselines** in our response to you). We will add them in our revised version. Thanks again for your time and consideration.
>
> Sincerely,
> The Authors

---

### Official Review · Reviewer_t7Fe · 2025-06-30

**Clarity:** 1
**Significance:** 2
**Originality:** 2
**Rating:** 4
**Confidence:** 2

**Summary:**

The authors provide a technique for re-training SNNs on image data at test-time to achieve higher accuracies on those samples.

**Questions:**

- I'm not an expert at test-time adaption, but I'm somewhat confused why this is needed. I understand SNNs would perform bad at test time if trained without data augmentation, and then performing the author's similarity aligning approach at test-time seems to help. But wouldn't it also help/solve the problem to just train with good data augmentation to begin with? I can't seem to find any controls in the paper to this extent that show this: train just on dataset and train on dataset with good augmentation and performing the author's test-time method.
- It's also unclear to which degree it's possible to perform the proposed algorithm on neuromorphic computers e.g. I'm not aware of neuromorphic performing batch data augmentation?
- Why was rate-coding chosen for the similarity measure - in particular as the sparsity of individual spikes is a nice property of SNNs when deploying them for inference on neuromorphic computers; shouldn't this be taken into account?

**Ethical Concerns:**

["NO or VERY MINOR ethics concerns only"]

**Final Justification:**

The author's have addressed my questions and concerns.

**Limitations:**

yes

**Paper Formatting Concerns:**

No - although line 249 has an erroneous "(".

**Quality:**

3

**Strengths And Weaknesses:**

Strengths:
- I'm impressed by the number of datasets and control models the authors test. Clearly they have put in a lot of effort into the work, which deserves merit.

Weaknesses:
- I'm a bit weary of the method actually performing better. The results in Table 1 would need statistical validation, I'd suggest running a Wilcoxon Signed-Rank Test.

---

> ### Author Rebuttal · Authors · 2025-07-30
>
> Thank you for your thoughtful comments, which have helped us make our work clearer and more solid. We have carefully addressed all your concerns in detail.
>
> **[W1] Wilcoxon Signed-Rank Test.** Thank you for your insightful suggestion. Initially, the results in Table 1 were obtained by running experiments three times with different random seeds and selecting the best outcome. Following your recommendation, we conducted two additional sets of experiments on CIFAR-10-C and performed a Wilcoxon Signed-Rank Test. The results shown in the table below, consistent with our original findings, confirm that our method achieves statistically significant improvements over the baselines. This further validates the effectiveness of our approach. We will give the Wilcoxon Signed-Rank Test for all experiments in the revised version.
> |Compared Method|Metric||Noise|||Blur||||Weather||||Digital||||
> |:----------------:|:------:|:----------:|:--------:|:---------:|:----------:|:---------:|:----------:|:--------:|:----------:|:-------:|:---------:|:---------:|:----------:|:---------:|:--------:|:--------:|:--------:|
> ||| **Gauss.** | **Shot** | **Impl.** | **Defoc.** | **Glass** | **Motion** | **Zoom** |  **Snow**  | **Fog** | **Frost** | **Brit.** | **Contr.** | **Elas.** | **Pix.** | **JPEG** | **Avg.** |
> | **vs.&nbsp;No&nbsp;Adapt** |  **W** |15|15|15|15|15|15|15|15|15|15|15|15|15|15|0|15|
> ||  **p** |0.0312|0.0312|0.0312|0.0312|0.0312|0.0312|0.0312|0.0312|0.0312|0.0312|0.0312|0.0312|0.0312|0.0312|1.0000|0.0312|
> |   **vs. SITA**   |  **W** |15|15|15|15|15|15|15|15|15|15|15|15|15|15|0|15|
> ||  **p** |0.0312|0.0312|0.0312|0.0312|0.0312|0.0312|0.0312|0.0312|0.0312|0.0312|0.0312|0.0312|0.0312|0.0312|1.0000|0.0312|
> |   **vs. MEMO**   |  **W** |15|0|15|15|15|15|15|15|15|15|10|9|0|15|0|15|
> ||  **p** |0.0312|1.0000|0.0312|0.0312|0.0312|0.0312|0.0312|0.0312|0.0312|0.0312|0.4000|0.3333|0.0312|0.0312|1.0000|0.0312|
>
> **[Q1] Explanation of Domain Shift.** Thank you for raising this interesting point. To clarify, our method addresses the challenge of domain shift, where the distribution of the test data differs from that of the training data. While training with data augmentation can improve generalization within **the same domain**, it cannot fully prepare the model for unseen domains, as training-time augmentations are inherently limited to the distributions observed or assumed during training. TTA, on the other hand, is specifically designed to handle **cross-domain** scenarios. In real-world applications, the domain shift caused by unknown factors, such as changes in environment or sensor characteristics, is often unpredictable and cannot be accounted for during training. Our method leverages the test-time data itself by performing augmentations within the shifted domain and aligning the features in our space, enabling unsupervised TTA to adapt effectively to the new domain, even when both labels and domain-shifted information are unavailable beforehand.
>
> **[Q2] Feasibility of Deployment.** On neuromorphic platforms like BrainScale, SpiNNaker2, and Loihi2, the data preprocessing unit and neural network inference module are typically **separate**. Batch data augmentation can be performed in the preprocessing stage, outside the neuromorphic core, before feeding data into the system for inference. This design allows augmentation to be handled efficiently without impacting the neuromorphic computing core, making our method feasible on such platforms.
>
> **[Q3] Similarity Measurement Objective.** Our method calculates the total spike count at only the final timestep and only focuses on one specific layer of neurons. This design choice ensures that it does not interfere with the sparse spike propagation characteristic of SNNs during inference. Additionally, as discussed in Appendix B, we explored aligning spikes at every timestep. However, this approach caused the model to overfit to augmentation-specific details, losing essential underlying information and ultimately degrading TTA performance. Our choice strikes a balance between preserving sparsity and achieving robust TTA.

---

> > ### Comment · Reviewer_t7Fe · 2025-08-05
> >
> > Thank you for answer, I have some follow up questions:
> >
> > [W1] How many samples where used (where every sample is network trained with different random weight init). It would also be insightful to have the test performed over the mean test accuracy of each class i.e. to have a final p-value per method over all augmentation methods within a dataset. Also, I'd like to see the statistical test performed on the other datasets e.g. CIFAR-100-C, SNN-VGG11.
> >
> > [Q3] Did you explore smoothing the spike trains and aligning in time as opposed to aligning spikes at every timestep? This might alleviate overfitting.

---

> ### Author Response · Authors · 2025-08-05
> **Thank You, Additional Experimental Results, and Clarifications**
>
> **[W1]**
> 1. The reported results are based on **5** independent runs.
>
> 2. Thanks for your advise. We performed a Wilcoxon signed-rank test on the per-class accuracies for CIFAR-10-C. Due to character limits, we present results for three representative corruption types below. We will add the full table covering all 15 corruptions to the appendix in our revised paper.
>
>     Gassian Noise:
>     |Compared Method|Metric|Class||||||||||
>     |-|-|-|-|-|-|-|-|-|-|-|-|
>     |||0|1|2|3|4|5|6|7|8|9|
>     |**vs.&nbsp;No&nbsp;Adapt**|**W**|15|15|15|15|15|15|15|15|15|15|
>     ||**p**|0.0312|0.0312|0.0312|0.0312|0.0312|0.0312|0.0312|0.0312|0.0312|0.0312|
>     |**vs. SITA**|**W**|15|15|15|15|15|15|15|15|15|15|
>     ||**p**|0.0312|0.0312|0.0312|0.0312|0.0312|0.0312|0.0312|0.0312|0.0312|0.0312|
>     |**vs. MEMO**|**W**|15|15|15|15|15|15|15|15|15|15|
>     ||**p**|0.0312|0.0312|0.0312|0.0312|0.0312|0.0312|0.0312|0.0312|0.0312|0.0312|
>
>     Shot Noise:
>     |Compared Method|Metric|Class||||||||||
>     |-|-|-|-|-|-|-|-|-|-|-|-|
>     |||0|1|2|3|4|5|6|7|8|9|
>     |**vs.&nbsp;No&nbsp;Adapt**|**W**|15|15|15|15|15|15|15|15|15|15|
>     ||**p**|0.0312|0.0312|0.0312|0.0312|0.0312|0.0312|0.0312|0.0312|0.0312|0.0312|
>     |**vs. SITA**|**W**|15|15|15|15|15|15|15|14|15|15|
>     ||**p**|0.0312|0.0312|0.0312|0.0312|0.0312|0.0312|0.0312|0.0625|0.0312|0.0312|
>     |**vs. MEMO**|**W**|5|0|0|5|5|0|0|0|14|0|
>     ||**p**|0.7812|1.0000|1.0000|0.7812|0.7812|1.0000|1.0000|1.0000|0.0625|1.0000|
>
>     Brightness:
>     |Compared Method|Metric|Class||||||||||
>     |-|-|-|-|-|-|-|-|-|-|-|-|
>     |||0|1|2|3|4|5|6|7|8|9|
>     |**vs.&nbsp;No&nbsp;Adapt**|**W**|15|15|15|15|15|15|15|15|15|15|
>     ||**p**|0.0312|0.0312|0.0312|0.0312|0.0312|0.0312|0.0312|0.0312|0.0312|0.0312|
>     |**vs. SITA**|**W**|15|15|15|15|15|15|15|15|15|15|
>     ||**p**|0.0312|0.0312|0.0312|0.0312|0.0312|0.0312|0.0312|0.0312|0.0312|0.0312|
>     |**vs. MEMO**|**W**|15|15|12|15|15|15|14|15|12|15|
>     ||**p**|0.0312|0.0312|0.1562|0.0312|0.0312|0.0312|0.0625|0.0312|0.1562|0.0312|
>
>     To validate these improvements, paired t-tests performed across all 15 corruption types confirm that our method's performance gains are statistically significant compared to all methods (p < 0.05):
>
>     |Comparison|p|
>     |-|-|
>     |**vs.&nbsp;No&nbsp;Adapt**|0.0312|
>     |**vs. SITA**|0.0312|
>     |**vs. MEMO**|0.0312|
>
> 3. We have extended this same statistical analysis to CIFAR-100-C with SNN-VGG11, which is shown below (the last column refers to the final p-value) and indicates its statistically significance:
>
>     |Compared Method|Metric|Noise|||Blur||||Weather||||Digital||||Acc.|
>     |-|-|-|-|-|-|-|-|-|-|-|-|-|-|-|-|-|-|
>     |||**Gauss.**|**Shot**|**Impl.**|**Defoc.**|**Glass**|**Motion**|**Zoom**|**Snow**|**Fog**|**Frost**|**Brit.**|**Contr.**|**Elas.**|**Pix.**|**JPEG**|**Avg.**|
>     |**vs.&nbsp;No&nbsp;Adapt**|**W**|15|15|15|15|15|15|15|15|15|15|15|15|15|15|15|15|
>     ||**p**|0.0312|0.0312|0.0312|0.0312|0.0312|0.0312|0.0312|0.0312|0.0312|0.0312|0.0312|0.0312|0.0312|0.0312|0.0312|**0.0312**|
>     |**vs. SITA**|**W**|15|15|15|15|15|15|15|15|15|15|15|10|15|15|15|15|
>     ||**p**|0.0312|0.0312|0.0312|0.0312|0.0312|0.0312|0.0312|0.0312|0.0312|0.0312|0.0312|0.0339|0.0312|0.0312|0.0312|**0.0312**|
>     |**vs. MEMO**|**W**|15|15|15|15|14|14|15|15|15|15|13|12|15|15|1|15|
>     ||**p**|0.0312|0.0312|0.0312|0.0312|0.0625|0.0625|0.0312|0.0312|0.0312|0.0312|0.0938|0.1562|0.0312|0.0312|0.9688|**0.0312**|
>
> **[Q3]** \
> Thank you for the insightful suggestion. We did explore Gaussian kernel smoothing to mitigate overfitting. However, its performance was suboptimal, likely due to information loss during smoothing. While both spike counts and smoothed spike trains involve information loss, their underlying principles differ. We think the key factor is not the loss itself but the quality and stability of the learning signal each representation provides for TTA.
>
> 1. **Spike Counts Create a Stable, Macro-Level Target**: Aligning spike counts is asking the model to ensure the same neurons fire with the same overall intensity across augmented views. This creates a robust loss landscape. A spike jittering in time does not change the loss, so the gradient provides a clear, consistent directive: "make this neuron more (or less) excitable for this input." It's a stable, macro-level adjustment that aligns the core feature activation.
>
> 2. **Smoothed Sequences Create a Conflicting, Micro-Level Target**: Aligning smoothed sequences is asking the model to match the exact temporal shape of the activity. This creates a noisy and conflicting loss landscape due to benign temporal jitter. For example, a spike occurring at t=10 in one view and t=12 in another generates contradictory gradients during optimization, as the model attempts to adjust the spike timings to align. This conflict can lead to suboptimal parameter updates, reducing the model's effectiveness across views and potentially degrading the quality of the learned representations.

---

> > ### Comment · Reviewer_t7Fe · 2025-08-06
> >
> > Thank you for addressing my concerns and questions, I have raised my score.

---

> > > ### Author Response · Authors · 2025-08-07
> > > **Gratitude for Your Constructive Feedback**
> > >
> > > Dear Reviewer t7Fe,
> > >
> > > Thanks for your time and consideration. We really appreciate your insightful comments and suggestions, which have significantly improved the completeness and persuasiveness of our work.
> > >
> > > Sincerely,
> > >
> > > Authors

---

### Official Review · Reviewer_sQFP · 2025-07-03

**Clarity:** 3
**Significance:** 2
**Originality:** 3
**Rating:** 5
**Confidence:** 3

**Summary:**

The paper identifies the challenges that traditional Test-Time Adaptation (TTA) methods face when applied to SNNs due to ignoring the fundamental essence of SNNs. For example, MEMO [4], a traditional TTA method, just focuses on optimizing the entropy of prediction and fails to capture the fine-grained spiking dynamics that encode essential features. Also, in real-world scenarios, access to source data is restricted due to sensitive privacy. To mitigate these issues, the paper proposes SPACE framework, the first source-free and single test-time adaptation framework. SPACE leverages the unique dynamic of SNNs to maximize the consistency of feature maps across several augmented versions of the same input sample. To this end, the paper conducts extensive experiments through several datasets to demonstrate the effectiveness and robustness of the proposed framework.

**Questions:**

1. Could the authors conduct more experiments to compare with the current or SOTA Source-free Domain Adaptation methods such as SHOT [1], RoTTA [2], and Tast [3],...


2. The authors argue that when the feature maps of augmented samples are highly similar, the model becomes less sensitive to variations at test time. Why does the model ensure that it will learn core characteristics of samples instead of noise or variation? How could the model do that?

3. Could the authors clarify why MMD did not work?

4. Could the authors provide additional theoretical guarantees for the model’s robustness?

5. Although the authors provide the impact of the number of augmentations on the model, could the authors investigate the model’s sensitivity to augmentation strength?

6. As the authors claims that their SNNs have the advantage of energy efficiency, could the authors conduct additional experiments for energy consumption, either theoretical estimates or real hardware profiling?

**Ethical Concerns:**

["NO or VERY MINOR ethics concerns only"]

**Final Justification:**

My concerns have been satisfactorily addressed.

**Limitations:**

1. The number of compared methods is limited, resulting in not highlighting the robustness of the proposed method comprehensively.

**Quality:**

3

**Strengths And Weaknesses:**

Strengths:

1. The paper is well-written and easy to follow.

2. The challenges when applying traditional TTA for SNNs are thoroughly discussed.

3. This paper is the first attempt to adopt both Source-free and Test-Time Adaptation to SNNs.

4. The motivation of the proposed method is reasonably discussed.

5. The proposed framework is rigorously evaluated across diverse architectures and real-world scenarios.



Weaknesses:

1. Generating many augmentations and doing forward/backward processes may introduce high-cost energy and memory, even becoming more severe when employed in real-time or low-energy neuromorphic hardware.

2. The authors claim that consistency-based feature alignment helps the model become more robust, but do not provide a theoretical analysis on robustness bounds.

3. The number of compared methods is limited; SITA and MEMO were published since 2022. It would be better to compare with the current or SOTA Source-free Domain Adaptation methods.

---

> ### Author Rebuttal · Authors · 2025-07-30
>
> Thank you for your thoughtful comments, which have helped us make our work clearer and more solid. We have carefully addressed all your concerns in detail.
>
> **[W1\&Q6] Analyses of Computational Cost.** To evaluate computational efficiency, we compared the GFLOPs of different baselines with the same augmented batch size, and SPACE achieved the lowest GFLOPs, demonstrating its efficiency. Regarding the additional cost of optimizing spike-behavior consistency, this step is performed only **once** per input in single-sample setting, without requiring whole model backpropagation or iterative updates. While real hardware profiling would provide deeper insights, SPACE is feasible for deployment on neuromorphic platforms like BrainScale, SpiNNaker2, and Loihi2, where the preprocessing unit and inference module are typically **separate**. Augmentation batches can be pre-generated during the preprocessing stage, outside the neuromorphic core, and then fed into the system for efficient inference. This separation ensures that augmentation does not burden the neuromorphic core, maintaining energy and memory efficiency while leveraging our single-sample TTA strategy.
> |Method|RoTTA|DeYO|SITA|MEMO|SPACE|
> |-|-|-|-|-|-|
> |GFLOPs|160.67|318.07|161.48|161.48|**158.11**|
> |Accuracy|67.79%|67.99%|66.41%|69.20%|**71.03%**|
>
> **[W2\&Q4] Theoretical analysis on Feature Alignment.** Our method is theoretically grounded in the well-established principle of **consistency regularization** [1,2], operationalized at inference time. Specifically, we generate an augmented batch from a single test sample and adapt the model by maximizing the similarity of its internal feature representations (i.e., spike counts). This enforces feature-level consistency, compelling the model to learn an invariant mapping for semantic-preserving transformations introduced by augmentations. Importantly, these augmentations remain **within a reasonable range (e.g., AugMix [9])**, ensuring robustness to meaningful, controlled perturbations without introducing excessive noise. \
> The robustness of our method can be understood from two complementary theoretical perspectives. From an information-theoretic viewpoint, the process approximates the objective of an information bottleneck by making the feature representations maximally informative about the input’s identity while minimizing sensitivity to augmentation-induced variations. This results in a more disentangled and robust feature space. From a geometric perspective, if test-time perturbations push a sample off the learned data manifold, our adaptation process effectively projects it back toward a stable, central point within the local manifold defined by the augmentations. This "denoising" effect mitigates the impact of perturbations, stabilizing the feature representation and ensuring reliable predictions. Thus, the enhanced robustness of our method arises naturally from enforcing invariance and representational stability within the SNN-TTA framework.
>
> **[W3\&Q1] Additional Baselines.**  We would like to clarify the setting of our work. In our **single-sample TTA** scenario, each sample undergoes test-time adaptation independently. This setting reflects real-world scenarios where it is often impractical to test with a mini-batch of data. Consequently, many batch-based TTA methods that rely on adjusting normalization layers [1,2] are not suitable for this setting. Moreover, some SNN architectures lack normalization layers altogether. Therefore, we initially chose SITA and MEMO, two single-instance TTA methods, as baselines. \
> For the ANN-based TTA methods such as SHOT[5] and TAST[6], they are incompatible with SNNs for the following reasons:
> 1. SHOT relies on pseudo-labeling in a static feature vector space, which disregards the critical temporal dynamics of SNNs. This results in incomplete feature representation and invalidates distance-based metrics.
> 2. SHOT freezes the static classifier layer, but SNN classification is tightly coupled with spiking dynamics over time. There is no standalone, static classification layer that can be frozen without disrupting the functionality.
> 3. TAST depends on prior probabilities, which are similarly unsuitable given SNNs' temporal and discrete spiking nature.
>
> To address the your suggestion, we additionally included RoTTA[7] and the recent DeYO[8] as baselines. Since both methods require batch data, we applied the same augmentation strategy as SPACE to create a pseudo-batch for each sample to ensure a fair comparison. The results are shown below. RoTTA's memory bank cannot be effectively utilized in the independent single-sample setting, leading to limited performance gains. DeYO's reliance on smooth probability outputs for confidence comparison does not align with SNN's discrete, spike-based outputs, introducing significant errors. Our SPACE method, designed specifically for SNNs, outperforms these baselines while maintaining lower computational cost.
> |Method||Noise|||Blur||||Weather||||Digital|||Acc.
> |-|-|-|-|-|-|-|-|-|-|-|-|-|-|-|-|-|
> ||**Gauss.**|**Shot**|**Impl.**|**Defoc.**|**Glass**|**Motion**|**Zoom**|**Snow**|**Fog**|**Frost**|**Brit.**|**Contr.**|**Elas.**|**Pix.**|**JPEG**|**Avg.**|
> |No&nbsp;Adapt|72.38%|74.70%|58.57%|63.05%|63.96%|64.44%|71.33%|76.32%|43.57%|75.72%|82.44%|22.54%|75.01%|70.25%|84.28%|66.57%|
> |RoTTA|73.60%|75.49%|61.61%|66.08%|67.53%|67.75%|72.40%|74.34%|51.54%|76.35%|80.98% |23.01%|74.23%|68.60%|83.29%|67.79%|
> |DeYO|73.97%|76.16%|62.45%|66.29%|67.37%|67.60%|73.52%|74.90%|50.68%|76.23%|81.30% |20.90%|75.19%|70.44%|82.85%|67.99%|
> |SITA|73.06%|74.15%|58.41%|62.94%|64.21%|64.36%|70.72%|76.67%|43.40%|75.24%|82.51% |22.02%|74.74%|69.62%|84.16%|66.41%|
> |MEMO|77.73%|79.50%|65.74%|65.61%|67.50%|66.24%|72.61%|77.38%|45.47%|78.64%|83.05% |23.83%|76.45%|73.25%|85.05%|69.20%|
> |SPACE|77.98%|79.34%|69.41%|71.59%|67.76%|72.14%|74.67%|78.43%|52.80%|79.59%|83.22% |23.85%|75.49%|76.24%|82.88%|71.03%|
>
> **[W2\&Q2\&Q5] Analysis of Augmentation Robustness Bounds.** While our work does not provide a formal theoretical analysis on robustness bounds, it builds upon the theoretical insights provided by AugMix [9], which is specifically designed to improve robustness and uncertainty calibration by enforcing consistent embeddings of augmented images. AugMix defines clear boundaries for transformations (e.g., maximum rotation angle of 30°, translation within ±16 pixels), ensuring that noise is not excessive and the model learns core, invariant characteristics instead of overfitting to spurious variations. AugMix provides 10 predefined strength levels, which control the intensity of augmentations applied. Our main setup followed MEMO's configuration (strength = 1), which is the lowest and most controlled level, ensuring robustness without introducing excessive noise. To investigate the model's sensitivity to augmentation strength, we conducted additional experiments with different AugMix strength. Within AugMix’s designed boundaries, accuracy remained stable, with only slight declines as strength increased. However, when we broke AugMix’s boundaries (e.g., increasing rotation angles to 50° or translation to ±18 pixels), accuracy dropped significantly, demonstrating that overly strong augmentations introduce harmful noise. These findings validate AugMix’s reliability and SPACE’s ability to focus on core features while maintaining robustness under controlled augmentation settings.
> |AugMix Strength|1|3|5|7|10|Larger Boundary|
> |-|-|-|-|-|-|-|
> |Acc. on Gaussian Noise|77.98%|77.88%|77.48%|77.41%|77.02%|74.52%|
>
> **[Q3] Why MMD did not work.** The MMD-based method still achieves good TTA performance overall. However, compared to the simpler approach of directly aligning spike counts without the MMD mapping, it does not provide further improvement and, in some cases, results in a slight drop in accuracy. We hypothesize that this is because the spike count feature map in its original space already captures the essential characteristics of the sample, and further mapping it into a higher-dimensional space introduces unnecessary complexity or noise that does not contribute to better alignment. MMD is generally effective when there is a large discrepancy between distributions, but in our case, the augmentations are controlled (e.g., via AugMix) and already maintain strong consistency. Thus, the direct alignment of spike counts is sufficient and avoids additional computational overhead. These findings suggest that our simpler approach is more effective and efficient for aligning spike behaviors in SNNs.
>
> Reference \
> [1] Tarvainen, Antti, and Harri Valpola. "Mean teachers are better role models: Weight-averaged consistency targets improve semi-supervised deep learning results." Advances in neural information processing systems 30 (2017). \
> [2] Zhang, Marvin, Sergey Levine, and Chelsea Finn. "Memo: Test time robustness via adaptation and augmentation." Advances in neural information processing systems 35 (2022): 38629-38642. \
> [3] Boudiaf, Malik, et al. "Parameter-free online test-time adaptation." CVPR, 2022. \
> [4] Lim, Hyesu, et al. "TTN: A Domain-Shift Aware Batch Normalization in Test-Time Adaptation." ICLR, 2023. \
> [5] Liang, Jian, Dapeng Hu, and Jiashi Feng. "Do we really need to access the source data? source hypothesis transfer for unsupervised domain adaptation." ICML, 2020. \
> [6] Jang, Minguk, Sae-Young Chung, and Hye Won Chung. "Test-Time Adaptation via Self-Training with Nearest Neighbor Information." ICLR, 2023. \
> [7] Yuan, Longhui, Binhui Xie, and Shuang Li. "Robust test-time adaptation in dynamic scenarios." CVPR, 2023. \
> [8] Lee, Jonghyun, et al. "ENTROPY IS NOT ENOUGH FOR TEST-TIME ADAPTATION: FROM THE PERSPECTIVE OF DISENTANGLED FACTORS." ICLR, 2024. \
> [9] Hendrycks, Dan, et al. "AugMix: A Simple Data Processing Method to Improve Robustness and Uncertainty." ICLR, 2020.

---

> ### Author Response · Authors · 2025-08-08
> **Follow-up on our rebuttal**
>
> Dear Reviewer sQFP,
>
> Thank you for your time and valuable feedback on our manuscript. We have posted our response to all reviews and would be grateful if you had a moment to take a look. **Please let us know if you have any further questions or concerns about our responses. We would be glad to address them (if applicable) as soon as possible**.
>
> Once again, thanks for your time and consideration. We are looking forward to your feedback.
>
> Sincerely,
>
> The Authors

---

> ### Comment · Reviewer_sQFP · 2025-08-09
>
> Dear authors,
>
> Thanks for your detailed rebuttal. The responses have addressed my concerns and I have no further questions. I will also raise my score.

---

> > ### Author Response · Authors · 2025-08-09
> > **Gratitude for Your Constructive Feedback**
> >
> > Dear Reviewer sQFP,
> >
> > Thanks for your time and consideration. We really appreciate your insightful comments and suggestions, which have significantly improved the completeness and persuasiveness of our work.
> >
> > Sincerely,
> >
> > Authors

---

### Official Review · Reviewer_hnfy · 2025-07-05

**Clarity:** 2
**Significance:** 1
**Originality:** 3
**Rating:** 2
**Confidence:** 4

**Summary:**

This work addresses domain shift by leveraging SNNs' unique spiking dynamics rather than output probabilities or batch normalization. The key insight is a spike-behavior consistency loss: For a single test sample and its augmented views, SPACE enforces similarity in local spike-based feature maps across the network layers. This approach bypasses limitations of existing methods that fail on SNNs due to their neglect of temporal spiking patterns and lack of BN layers.

**Questions:**

See weakness

**Ethical Concerns:**

["NO or VERY MINOR ethics concerns only"]

**Limitations:**

The focus on domain adaptation overlooks SNNs' more fundamental challenge: closing the accuracy gap with ANNs. The practical value of TTA for current SNNs warrants scrutiny.

**Quality:**

2

**Strengths And Weaknesses:**

**Strengths: **

S1: Directly exploits spike dynamics as adaptation signals—aligning with SNNs' fundamental computation mechanism. This addresses a critical gap in TTA literature, which predominantly targets ANNs.

S2: Eliminates privacy/bandwidth constraints by avoiding source data.

S3: Works with modern SNN architectures lacking BatchNorm.


** Weaknesses: **

W1: While SPACE leverages spike dynamics, the computational cost of optimizing spike-behavior consistency is not explicitly analyzed. For SNNs, which are often prized for low-latency inference, this overhead could affect practical deployment.

W2: The impact of spike encoding methods (e.g., rate vs. temporal coding) on adaptation efficacy remains unexplored. Theoretical grounding for encoding-dependent robustness is needed.

W3: The influence of critical parameters like time steps on adaptation performance is not discussed, despite their direct link to SNNs' temporal feature extraction.

W4: Equation (4) relies on converting spike trains to continuous values for channel-wise alignment. Implementation details and potential information loss during conversion require clarification.

W5: The focus on domain adaptation overlooks SNNs' more fundamental challenge: closing the accuracy gap with ANNs. The practical value of TTA for current SNNs warrants scrutiny.

---

> ### Author Rebuttal · Authors · 2025-07-30
>
> Thank you for your thoughtful comments, which have helped us make our work clearer and more solid. We have carefully addressed all your concerns in detail.
>
> **[W1] Analyses of Computational Cost.** To evaluate computational efficiency, we compared the GFLOPs of different baselines with the same augmented batch size, and SPACE achieved the lowest GFLOPs, demonstrating its efficiency. Regarding the additional cost of optimizing spike-behavior consistency, this step is performed only **once** per input in single-sample setting, without requiring whole model backpropagation or iterative updates. While real hardware profiling would provide deeper insights, SPACE is feasible for deployment on neuromorphic platforms like BrainScale, SpiNNaker2, and Loihi2, where the preprocessing unit and inference module are typically **separate**. Augmentation batches can be pre-generated during the preprocessing stage, outside the neuromorphic core, and then fed into the system for efficient inference. This separation ensures that augmentation does not burden the neuromorphic core, maintaining energy and memory efficiency while leveraging our single-sample TTA strategy.
> |Method|RoTTA|DeYO|SITA|MEMO|SPACE|
> |-|-|-|-|-|-|
> |GFLOPs|160.67|318.07|161.48|161.48|**158.11**|
> |Accuracy|67.79%|67.99%|66.41%|69.20%|**71.03%**|
>
> **[W2] Effectiveness on Temporal Coding Model.** We thank the reviewer for this insightful suggestion. To explore this, we conducted additional experiments by replacing the Poisson rate coding in SNN-VGG9 with temporal coding—where neurons representing higher pixel values fire spikes earlier—while keeping all other structures and hyperparameters identical during training. We first observed that the baseline temporal-coded model is inherently less robust, showing a larger accuracy drop on domain-shifted data compared to its rate-coded counterpart. We posit this is because temporal coding encodes information in precise spike timings, making it highly sensitive to input perturbations that can disproportionately alter these timings. In contrast, rate coding relies on a statistical average of spikes, providing natural redundancy and resilience to such noise.
> |Coding|Clean Data|Shifted Data|Acc. Loss|
> |-|-|-|-|
> |Temporal Coding|89.27%|58.11%|**31.16%**|
> |Rate Coding|90.61%|66.57%|**24.04%**|
>
> While our method measures total spike counts rather than their precise timing, its enhanced efficacy in temporal-coded models stems from a powerful underlying principle: the total spike count serves as a highly sensitive proxy for temporal stability due to **the temporal boundary effect**. The core insight is that we regularize the macroscopic outcome (the count) to stabilize the microscopic cause (the timing). In a temporal coding regime, minor input perturbations cause significant spike timing jitter. For any spike occurring near the edge of the fixed processing window, even slight jitter can push its timing across this boundary. This induces a discrete, "all-or-nothing" change in that neuron's contribution to the total count (e.g., from 1 to 0). Our TTA method, by enforcing consistency in the total spike count across augmentations, directly targets the cumulative result of these boundary-crossing events. To satisfy this objective, the model is implicitly forced to adapt its weights to suppress the underlying temporal jitter, ensuring spike timings are generated robustly and do not erratically cross the window boundary. This discrete, high-magnitude error signal is far more pronounced than the small statistical fluctuations in a rate-coded model, allowing our method to apply a stronger and more effective corrective pressure, thus yielding a greater improvement in robustness. \
> Crucially, this approach provides a significant advantage: by operating on the aggregate count rather than enforcing a strict, spike-for-spike temporal alignment, our method avoids the dual risks of overfitting to the specific augmentations and catastrophically forgetting the model's vital pre-trained knowledge.
> |Method||Noise|||Blur||||Weather||||Digital|||Acc.
> |-|-|-|-|-|-|-|-|-|-|-|-|-|-|-|-|-|
> ||**Gauss.**|**Shot**|**Impl.**|**Defoc.**|**Glass**|**Motion**|**Zoom**|**Snow**|**Fog**|**Frost**|**Brit.**|**Contr.**|**Elas.**|**Pix.**|**JPEG**|**Avg.**|
> |Temporal&nbsp;Coding|||||||
> |No Adapt|39.74%|39.32%|68.42%|59.59%|54.43%|60.19%|65.90%|69.78%|46.21%|66.77%|78.14%|27.26%|70.29%|46.00%| 79.59%|58.11%|
> |SPACE|45.19%|44.00%|71.82%|64.16%|56.31%|64.90%|68.65%|70.29%|56.90%|72.40%|79.09%|68.64%|70.66%|50.06%|80.07%|64.21%(**+6.10%**)|
> |Rate&nbsp;Coding|||||||||
> |No Adapt|72.38%|74.70%|58.57%|63.05%|63.96%|64.44%|71.33%|76.32%|43.57%|75.72%|82.44%|22.54%|75.01%|70.25%|84.28%|66.57%|
> |SPACE|77.98%|79.34%|69.41%|71.59%|67.76%|72.14%|74.67%|78.43%|52.80%|79.59%|83.22%|23.85%|75.49%|76.24%|82.88%|71.03%(**+4.46%**)|
>
> **[W3] Impact of Time Steps.** The time steps in our method are predetermined during the model's pretraining phase and remain fixed during TTA. Regardless of the specific number of inference time steps, our method operates by aggregating spike counts across the entire time window to extract temporal features. This design ensures that our approach is **independent** of the exact time step setting. In our experiments, we evaluated four different backbones with varying time steps: SNN-VGG (25), SNN-ResNet (30), Spike-driven Transformer V3 (4), and SNN-ConvLSTM (32). The consistent performance of our method across these diverse architectures demonstrates its robustness and generalizability, even when applied to models with different temporal configurations. This independence from time step settings highlights the flexibility of our TTA framework and its ability to adapt effectively across various SNN structures.
>
> **[W4] Potential Info Loss in Equation (4).** Equation (4) relies on converting discrete spike counts into continuous probability values via a softmax function. This transformation is not a source of unintended information loss, but rather a **principled mechanism for robust feature selection** that is central to our method's success. The "information loss" is the intentional discarding of the absolute magnitude of spike counts to focus on the **relative importance** of neurons within a channel. By applying softmax, we normalize away noisy fluctuations in absolute firing rates and create a stable spatial saliency map. Aligning these maps forces the model to learn which spatial features are consistently most important, directly encouraging the learning of invariant representations. \
> In addition, while our alignment objective operates on this spatially-focused abstraction, it simultaneously **preserves and refines the inherent temporal characteristics of the SNN**. The underlying model remains a fully temporal processor, and the spike counts are the direct result of its complex dynamics. By demanding consistency in the outcome (the count-based saliency map), our method implicitly forces the network to stabilize the cause (the spike generation process). This creates a powerful regularization pressure that reduces temporal jitter and enhances the robustness of spike timing, all without the computational cost and overfitting risks associated with direct, spike-for-spike temporal alignment.
>
> **[W5] Motivation of the Work.** Thank you for this thoughtful question, as it touches upon the core motivation for our research. We agree completely that the practical value of any new method is a paramount concern, and we appreciate the opportunity to clarify our perspective on why TTA is a crucial and timely area of study for SNNs. \
> Our work is built upon the exciting recent progress in the SNN field. While a performance gap has historically been a key consideration, several recent studies [1,2] have shown that SNNs can now achieve accuracy that is highly competitive with their ANN counterparts. This convergence in performance makes SNNs a highly compelling choice for applications where their unique architectural advantages are paramount. Specifically, for tasks involving event-based sensing (e.g., DVS cameras) or deployment on resource-constrained edge hardware, the inherent efficiency and temporal processing capabilities of SNNs make them a more natural and powerful solution than traditional ANNs. Crucially, in these target applications—robotics navigating new environments, devices operating in changing weather, or wearables adapting to user behavior—domain shift is not a hypothetical risk, but an operational certainty. An SNN that achieves high accuracy on a static benchmark but fails when faced with real-world variations is of limited practical utility. We therefore posit that **robustness to domain shift is a prerequisite for deployment, not a post-hoc optimization**. \
> Addressing this adaptation challenge, however, is not straightforward. The established TTA techniques developed for ANNs are often not directly transferable to the discrete, sparse, and temporal dynamics of spiking neurons (referring to my response to Reviewer sQFP [W3&Q1] and Reviewer 4hgi [W2]). This highlights a specific need for adaptation methods designed for the SNN paradigm. Therefore, our research is motivated by the goal of bridging this gap. By developing a TTA method tailored for SNNs, we aim to enhance their robustness and practical value in the very domains where they are already becoming a preferred solution. We see this as a necessary and complementary step to the ongoing work on accuracy, ensuring that high-performing SNNs are also reliable and effective when deployed in the dynamic conditions of the real world.
>
> Reference \
> [1] Yao, Man, et al. "Scaling spike-driven transformer with efficient spike firing approximation training." IEEE Transactions on Pattern Analysis and Machine Intelligence (2025). \
> [2] Zhou, Zhaokun, et al. "Spikformer: When Spiking Neural Network Meets Transformer." The Eleventh International Conference on Learning Representations, 2023.

---

### Author Response · Authors · 2025-08-09
**Clarification on the Motivation and Significance of Our Work**

Dear Program Chairs, Senior Area Chairs, Area Chairs and Reviewers,

Thank you for your valuable time and insightful feedback. We would like to take this opportunity to offer some clarifications on the motivation and significance of our work, particularly in response to key points raised during the review process.

---

* **Improving SNN robustness towards real-world deployment gains increasing interest from the community.** For example, recent NeurIPS publications such as [1], [2], and [3] have all investigated the crucial issues of generalization and robustness in SNNs.

* **Test-time adaptation of SNN is a more practical scheme for improving inference performance under unknown test-time corruptions.**
Notably, [1] focuses on Source-Free Domain Adaptation (SFDA), which operates in a batch-based manner, requiring updates at an "epoch's end" over a target dataset. Our work tackles a more challenging and dynamic scenario: **our TTA adapts to each incoming sample individually and "on-the-fly".** This removes the need for data pre-collection, making our approach more directly applicable to the autonomous, real-world deployments where SNNs are expected to excel.

* **Our proposed method also performs better than additional state-of-the-art ANN baselines.** In response to feedback about the initial scope of our baselines, we have now benchmarked our method against two additional state-of-the-art ANN methods (RoTTA and DeYO) in our revision. The new results further validate the effectiveness of our approach and comprehensively address this technical point.

---

We hope these clarifications underscore the importance of our research problem and the advantages of our proposed method. We firmly believe that **test-time adaptation of SNN is an urgent and practical roadmap towards the real-world deployment of current SNN models.**

Thank you for your time and consideration.

Sincerely,

Authors of Paper 15694

Reference\
[1] Guo, Weiyu, et al. "SpGesture: Source-Free Domain-adaptive sEMG-based Gesture Recognition with Jaccard Attentive Spiking Neural Network." Advances in Neural Information Processing Systems 37 (2024): 36717-36747.\
[2] Ding, Jianhao, et al. "Snn-rat: Robustness-enhanced spiking neural network through regularized adversarial training." Advances in Neural Information Processing Systems 35 (2022): 24780-24793.\
[3] Xu, Mengting, et al. "FEEL-SNN: Robust spiking neural networks with frequency encoding and evolutionary leak factor." Advances in Neural Information Processing Systems 37 (2024): 91930-91950.

---

### Note · Authors · 2025-08-11

Dear Program Chairs, Senior Area Chairs, Area Chairs and Reviewers,

Thank you for the constructive discussion. We would like to offer a final remarks of our contributions and clarifications.

Our work introduces the first TTA method specifically designed for SNNs. It not only establishes a new state-of-the-art for SNN robustness but also outperforms leading ANN-TTA methods while preserving the high efficiency inherent to SNNs.

In our rebuttal, we believe we have comprehensively addressed the reviewers' concerns. We strengthened our **theoretical analysis** of the feature alignment mechanism, added **extensive SOTA ANN baselines** to empirically validate our method's necessity and superior performance (confirmed by **statistical tests**), analyzed the **incompatibility** of ANN-TTA methods on SNNs, and clarified our method's practical **feasibility on neuromorphic hardware**.

The most critical remaining point is the work's motivation. We respectfully argue that focusing on SNN robustness is a timely and vital research direction for three key reasons:

---
* **The Performance Gap is Closing:** SOTA SNNs (refers to response @hnfy) have already achieved performance comparable to their ANN counterparts on clean data. This naturally shifts the research frontier towards real-world challenges like robustness, which is the direct focus of our work.
* **Community Interest is Growing:** Improving SNN robustness is a topic of increasing importance, as evidenced by recent NeurIPS publications ([1], [2], [3]) that investigate this very issue. Our work is squarely aligned with this critical research trend.
* **Our Paradigm is More Practical:** Our single-sample adaptation tackles a more challenging and realistic scenario for autonomous systems than existing batch-based domain adapatation method [1]. This directly addresses a key hurdle for deploying SNNs in dynamic, real-world environments.
---
We are confident that our work presents a significant and necessary step towards building truly robust and deployable SNNs. We thank you for your time and consideration.

Sincerely,

Authors of Paper 15694

Reference\

[1] SpGesture: Source-Free Domain-adaptive sEMG-based Gesture Recognition with Jaccard Attentive Spiking Neural Network, NeurIPS 2024. \
[2] Snn-rat: Robustness-enhanced spiking neural network through regularized adversarial training, NeurIPS 2022. \
[3] FEEL-SNN: Robust spiking neural networks with frequency encoding and evolutionary leak factor, NeurIPS 2024.

---

### Decision · Program_Chairs · 2025-09-17

**Decision:**

Accept (poster)

**Comment:**

## Summary of Scientific Claims and Findings
This paper proposes SPACE, the first test-time adaptation (TTA) algorithm specifically designed for Spiking Neural Networks (SNNs). It addresses SNNs’ vulnerability to domain shifts by leveraging spike-aware consistency regularization, aligning spike-based local feature maps across augmented views of single test samples without requiring source data. SPACE is evaluated on multiple datasets and diverse SNN architectures—including CNNs, Transformers, and ConvLSTMs—demonstrating superior robustness and performance compared to state-of-the-art ANN and SNN TTA baselines. The method is theoretically grounded in consistency regularization and information bottleneck principles and is argued to be practical for deployment on neuromorphic hardware.


## Strengths
- Novelty: The first source-free, single-instance TTA method tailored to the discrete, temporal, and event-driven nature of SNNs.
- Theoretical grounding: Provides both information-theoretic and geometric interpretations of feature alignment to justify robustness improvements.
- Comprehensive empirical evaluation: Extensive experiments across multiple datasets, architectures, and corruption types.
- Demonstrated superiority: Outperforms multiple state-of-the-art ANN and SNN TTA methods with statistical significance.
- Practicality: Discusses efficient implementation on neuromorphic platforms and addresses computational overhead concerns.
- Clear motivation: Argues convincingly for the urgency of robustness and adaptation in SNNs as they approach ANN-level accuracy.

##  Weaknesses
- Initial lack of comparison to recent state-of-the-art ANN TTA methods (RoTTA, DeYO) which was later addressed in rebuttal.
- Some theoretical aspects, such as formal robustness bounds, are not fully developed but partially supported by related literature (AugMix).
- Limited discussion on the sensitivity of the method to augmentation strength and hyperparameters in the initial submission.
- Some clarity issues regarding architectural details (e.g., definition of feature extractor vs. classifier, local vs. global feature maps) that were clarified post-review.

## Discussion and Rebuttal Summary

### Reviewers and Their Key Concerns:
- **Reviewer hnfy:**
  - Requested analysis of computational cost and clarification on practical deployment on neuromorphic hardware.
  - Asked about impact of spike encoding (rate vs. temporal coding) and time steps on adaptation efficacy.
  - Requested clarification on equation implementation details and motivation regarding SNN vs. ANN accuracy gap.

- **Reviewer sQFP:**
  - Suggested comparisons to additional state-of-the-art source-free domain adaptation methods (e.g., SHOT, RoTTA, TAST).
  - Questioned the theoretical guarantees on robustness and augmentation sensitivity.
  - Raised concerns about computational cost and energy efficiency in neuromorphic deployment.

- **Reviewer t7Fe:**
  - Requested statistical validation of results and further clarification on motivation for TTA over strong data augmentation at training.
  - Asked about smoothing spike trains for temporal alignment.

- **Reviewer 4hgi:**
  - Requested empirical support for SNNs’ vulnerability to domain shift relative to ANNs.
  - Noted initial lack of ANN baselines and clarity issues on the SPACE framework’s architectural components.
  - Asked about similarity function choices and kernel function details.

### Authors’ Responses:
- Provided detailed computational cost analysis, demonstrating SPACE’s efficiency and feasibility on neuromorphic platforms.
- Conducted new experiments analyzing spike encoding impacts and showing method’s robustness across different SNN time steps and architectures.
- Clarified feature alignment implementation, theory, and similarity metrics, emphasizing stability and avoidance of overfitting.
- Added comparisons to additional ANN TTA baselines (RoTTA, DeYO), showing SPACE’s superior performance and efficiency.
- Performed Wilcoxon signed-rank tests confirming statistical significance of improvements.
- Clarified architectural definitions of feature extractor and classifier, and local vs. global feature maps.
- Addressed motivation concerns by showing recent SNN-ANN accuracy convergence and necessity of adaptation for real-world deployment.
- Explained why smoothing spike trains leads to overfitting and why spike count alignment is more effective.
- Maintained responsiveness to reviewer feedback; reviewer sQFP raised their score after rebuttal, and others acknowledged clarifications while retaining borderline stances.

### Final Decision:
The authors substantially addressed reviewers’ major concerns with additional experiments, theoretical clarifications, and broader empirical baselines. The method’s novelty, practical importance, and strong empirical results justify acceptance. Remaining minor clarity and motivation points are addressed sufficiently in rebuttal. I recommend acceptance of this paper for its novel, well-motivated, and empirically strong contribution to the test-time adaptation of spiking neural networks, an important step toward their robust real-world deployment.